# On the Validity of Modeling SGD with Stochastic Differential Equations (SDEs)

**Zhiyuan Li   Sadhika Malladi   Sanjeev Arora**
Princeton University
{zhiyuanli,smalladi,arora}@cs.princeton.edu

## Abstract

It is generally recognized that finite learning rate (LR), in contrast to infinitesimal LR, is important for good generalization in real-life deep nets. Most attempted explanations propose approximating finite-LR SGD with Itô Stochastic Differential Equations (SDEs), but formal justification for this approximation (e.g., (Li et al., 2019a)) only applies to SGD with tiny LR. Experimental verification of the approximation appears computationally infeasible. The current paper clarifies the picture with the following contributions: (a) An efficient simulation algorithm SVAG that provably converges to the conventionally used Itô SDE approximation. (b) A theoretically motivated testable necessary condition for the SDE approximation and its most famous implication, the linear scaling rule (Goyal et al., 2017), to hold. (c) Experiments using this simulation to demonstrate that the previously proposed SDE approximation can meaningfully capture the training and generalization properties of common deep nets.

## 1   Introduction

Training with Stochastic Gradient Gescent (SGD) (1) and finite learning rate (LR) is largely considered essential for getting best performance out of deep nets: using infinitesimal LR (which turns the process into *Gradient Flow* (GF)) or finite LR with full gradients results in noticeably worse test error despite sometimes giving better training error (Wu et al., 2020; Smith et al., 2020; Bjorck et al., 2018).

Mathematical explorations of the implicit bias of finite-LR SGD toward good generalization have focused on the *noise* arising from gradients being estimated from small batches. This has motivated modeling SGD as a *stochastic process* and, in particular, studying Stochastic Differential Equations (SDEs) to understand the evolution of net parameters.

Early attempts to analyze the effect of noise try to model it as as a fixed Gaussian (Jastrzebski et al., 2017; Mandt et al., 2017). Current approaches approximate SGD using a parameter-dependent noise distribution that match the first and second order moments of of the SGD (Equation (2)). It is important to realize that this approximation is *heuristic* for finite LR, meaning it is not known whether the two trajectories actually track each other closely. Experimental verification seems difficult because simulating the (continuous) SDE requires full gradient/noise computation over suitably fine time intervals. Recently, Li et al. (2017, 2019a); Feng et al. (2017); Hu et al. (2019) provided rigorous proofs that the trajectories are arbitrarily close in a natural sense, but the proof needs the LR of SGD to be an unrealistically small (unspecified) constant so the approximation remains heuristic. In the worst case, the LR needs to be exponentially small, i.e., $e^{-\Omega(T)}$, where $T$ is the continuous training time. Furthermore, noise plays no role in these approximation analyses and the same analysis indeed shows GD, SGD and SDE all converge weakly to GF at the same rate. Thus whenever their requirements for LR are met, there should be no performance difference between SGD and full-batch GD. However, for a common practical LR choice we observe some difference in Figure 1, indicating that the LR is usually outside of the regime their result requires.

35th Conference on Neural Information Processing Systems (NeurIPS 2021).

Setting aside the issue of *correctness* of the SDE approximation, there is no doubt it has yielded important insights of practical importance, especially the *linear scaling rule* (LSR; see Definition 2.1) relating batch size and optimal LR, which allows much faster training using high parallelism (Krizhevsky, 2014; Goyal et al., 2017). However, since the scaling rule depends upon the validity of the SDE approximation, it is not mathematically understood when the rule fails. (Empirical investigation, with some intuition based upon analysis of simpler models, appears in (Goyal et al., 2017; Smith et al., 2020).

This paper casts new light on the SDE approximation via the following contributions:

1. A new and efficient numerical method, *Stochastic Variance Amplified Gradient (SVAG)*, to test if the trajectories of SGD and its corresponding SDE are close for a given model, dataset, and hyperparameter configuration. In Theorem 4.3, we prove (using ideas similar to Li et al. (2019a)) that SVAG provides an order-1 weak approximation to the corresponding SDE. (Section 4)

2. Empirical testing showing that the trajectory under SVAG converges and closely follows SGD, suggesting (in combination with the previous result) that the SDE approximation can be a meaningful approach to understanding the implicit bias of SGD in deep learning.

3. New theoretical insight into the observation in (Goyal et al., 2017; Smith et al., 2020) that linear scaling rule fails at large LR/batch sizes (Section 5). It applies to networks that use normalization layers (*scale-invariant* nets in Arora et al. (2019b)), which includes most popular architectures. We give a necessary condition for the SDE approximation to hold: *at equilibrium, the squared gradient norm must be smaller than its variance*.

## 2 Preliminaries and Overview

We use $|\cdot|$ to denote the $\ell_2$ norm of a vector and $\otimes$ to denote the tensor product. Stochastic Gradient Descent (SGD) is often used to solve optimization problems of the form $\min_{x \in \mathbb{R}^d} \mathcal{L}(x) := \mathbb{E}_\gamma \mathcal{L}_\gamma(x)$ where $\{\mathcal{L}_\gamma : \gamma \in \Gamma\}$ is a family of functions from $\mathbb{R}^d$ to $\mathbb{R}$ and $\gamma$ is a $\Gamma$-valued variable, e.g., denoting a random batch of training data. We consider the general case of an expectation over arbitrary index sets and distributions.

$$x_{k+1} = x_k - \eta \nabla \mathcal{L}_{\gamma_k}(x_k), \qquad \textit{(SGD)} \qquad (1)$$

where each $\gamma_k$ is an i.i.d. random variable with the same distribution as $\gamma$. Taking learning rate (LR) $\eta$ toward $0$ turns SGD into (deterministic) Gradient Descent (GD) with infinitesimal LR, also called *Gradient Flow*. Infinitesimal LR is more compatible with traditional calculus-based analyses, but SGD with finite LR yields the best generalization properties in practice. Stochastic processes give a way to (heuristically) model SGD as a continuous-time evolution (i.e., stochastic differential equation or SDE) without ignoring the crucial role of noise. Driven by the intuition that the benefit of SGD depends primarily on the covariance of noise in gradient estimation (and not, say, the higher moments), researchers arrived at following SDE for parameter vector $X_t$:

$$\mathrm{d}X_t = -\nabla \mathcal{L}(X_t)\mathrm{d}t + (\eta \Sigma(X_t))^{1/2}\mathrm{d}W_t \qquad \textit{(SDE approximation)} \qquad (2)$$

where $W_t$ is Wiener Process, and $\Sigma(X) := \mathbb{E}[(\nabla \mathcal{L}_\gamma(X) - \nabla \mathcal{L}(X))(\mathcal{L}_\gamma(X) - \nabla \mathcal{L}(X))^\top]$ is the covariance of the gradient noise. When the gradient noise is modeled by white noise as above, it is called an *Itô SDE*. Replacing $W_t$ with a more general distribution with stationary and independent increments (i.e., a *Lévy process*, described in Definition A.1) yields a *Lévy SDE*.

The SDE view—specifically, the belief in key role played by noise covariance—motivated the famous *Linear Scaling Rule*, a rule of thumb to train models with large minibatch sizes (e.g., in highly parallel architectures) by changing LR proportionately, thereby preserving the scale of the gradient noise.

**Definition 2.1** (Linear Scaling Rule (LSR)). (Krizhevsky, 2014; Goyal et al., 2017) When multiplying the minibatch size by $\kappa > 0$, multiply the learning rate (LR) also by $\kappa$.

If the SDE approximation accurately captures the SGD dynamics for a specific training setting, then LSR should work; however, LSR can work even when the SDE approximation fails. We hope to (1) understand when and why the SDE approximation can fail and (2) provide provable and practically applicable guidance on when LSR can fail. Experimentally verifying if the SDE approximation is

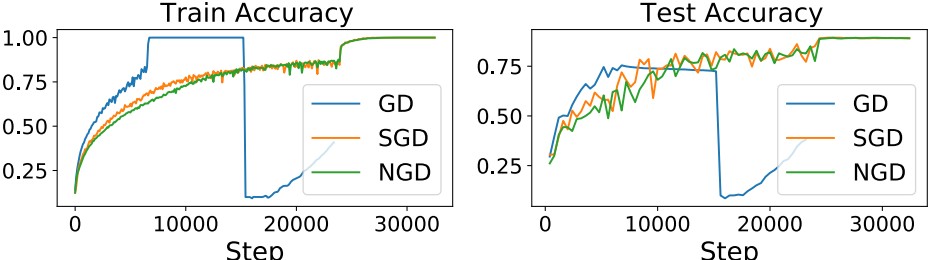

Figure 1: Non-Gaussian noise is not essential to SGD performance. SGD with batch size 125 and NGD with matching covariance have close train and test curves when training on CIFAR-10. $\eta = 0.8$ for all three settings and is decayed by 0.1 at step 24000. GD achieves 75.5% test accuracy, and SGD and NGD achieve 89.4% and 89.3%, respectively. We smooth the training curve by dividing it into intervals of 100 steps and recording the average. For efficient sampling of Gaussian noise, we use GroupNorm instead of BatchNorm and turn off data augmentation. The sudden drop of accuracy when using GD is not a coincidence, but a consequence of interplay between normalization and Weight Decay. See more discussion and implementation details in Appendix F.3.

valid is computationally challenging, because it requires repeatedly computing the full gradient and the noise covariance at very fine time intervals, e.g. the *Euler-Maruyama* method Equation (16). We are not aware of any empirical verification using conventional techniques, which we discuss in more detail in Appendix A.1. Section 4 gives a new, tractable simulation algorithm, SVAG, and presents theory and experiments suggesting it is a reasonably good approximation to both the SDE and SGD.

**Formalizing closeness of two stochastic processes.** Two stochastic processes (e.g., SGD and SDE) track each other closely if they lead to similar distributions on outcomes (e.g., trained nets). Mathematics formulates closeness of distributions in terms of expectations of suitable classes of test functions[1]; see Section 4.2. The test functions of greatest interest for ML are of course train and test error. These do not satisfy formal conditions such as differentiability assumed in classical theory but can be still used in experiments (see Figure 4). Section 5 uses test functions such as weight norm $|x_t|$, gradient norm $|\nabla\mathcal{L}(x_t)|$ and trace of noise covariance $\mathrm{Tr}[\Sigma(x_t)]$ and proves a sufficient condition for the failure of SDE approximation.

Mathematical analyses of closeness of SGD and SDE will often consider the discrete process

$$\hat{x}_{k+1} = \hat{x}_k - \eta\nabla\mathcal{L}(\hat{x}_k) + \eta\Sigma^{\frac{1}{2}}(\hat{x}_k)z_k, \qquad \textit{(Noisy Gradient Descent/NGD)} \qquad (3)$$

where $z_k \overset{\text{i.i.d.}}{\sim} N(0, I_d)$. A basic step in analysis will be the following *Error Decomposition*:

$$\mathbb{E}g(X_{\eta k}) - \mathbb{E}g(x_k) = \underbrace{(\mathbb{E}g(X_{\eta k})) - \mathbb{E}g(\hat{x}_k))}_{\text{Discretization Error}} + \underbrace{(\mathbb{E}g(\hat{x}_k) - \mathbb{E}g(x_k))}_{\text{Gap due to non-Gaussian noise}} \qquad (4)$$

**Understanding the failure caused by discretization error:** In Section 5, a testable condition of SDE approximation is derived for scale-invariant nets (i.e. nets using normalization layers). This condition only involves the *Noise-Signal-Ratio*, but not the shape of the noise. We further extend this condition to LSR and develops a method predicting the largest batch size at which LSR succeeds, which only takes a single run with small batch size.

## 2.1 Understanding the Role of Non-Gaussian Noise

Some works have challenged the traditional assumption that SGD noise is Gaussian. Simsekli et al. (2019); Nguyen et al. (2019) suggested that SGD noise is heavy-tailed, which Zhou et al. (2020) claimed causes adaptive gradient methods to generalize better than SGD. Xie et al. (2021) argued that the experimental evidence in (Simsekli et al., 2019) made strong assumptions on the nature of the gradient noise, and we furthermore prove in Appendix B.3 that their measurement method could flag Gaussian distributions as non-Gaussian. Below, we clarify how the Gaussian noise assumption interacts with our findings.

**Non-Gaussian noise is not essential to SGD performance.** We provide experimental evidence in Figure 1 and Appendix F.3 that SGD (1) and NGD (3) with matching covariances achieve

---

[1]The discriminator net in GANs is an example of test function in machine learning.

similar test performance on CIFAR10 ($\sim 89\%$), suggesting that even if the gradient noise in SGD is non-Gaussian, modeling it by a Gaussian estimation is sufficient to understand generalization properties. Similar experiments were conducted in (Wu et al., 2020) but used SGD with momentum and BatchNorm, which prevents the covariance of NGD noise from being equal to that of SGD. These findings confirm the conclusion in (Cheng et al., 2020) that differences in the third-and-higher moments in SGD noise don't affect the test accuracy significantly, though differences in the second moments do.

**LSR can work when SDE approximation fails.** We note that (Smith et al., 2020) derives LSR (Definition 2.1) by assuming the Itô SDE approximation (2) holds, but in fact the validity of the SDE approximation is a sufficient but not necessary condition for LSR to work. In Section B.1, we provide a concrete example where LSR holds for all LRs and batch sizes, but the dynamics are constantly away from the Itô SDE limit. This example also illustrates that the failure of the SDE approximation can be caused solely by non-Gaussian noise, even when there is no discretization error (i.e., the loss landscape and noise distribution are parameter-independent).

**SVAG does not require Gaussian gradient noise.** In Section 4, we present an efficient algorithm SVAG to simulate the Itô SDE corresponding to a given training setting. In particular, Theorem 4.3 reveals that SVAG simultaneously causes the discretization error and the gap by non-Gaussian noise to disappear as it converges to the SDE approximation. From Figure 4 and Appendix F.1, we can observe that for vision tasks, the test accuracy of deep nets trained by SGD in standard settings stays the same when interpolating towards SDE via SVAG, suggesting that neither the potentially non-Gaussian nature of SGD noise nor the discrete nature of SGD dynamics is an essential ingredient of the generalization mystery of deep learning.

# 3    Related Work

**Applications of the SDE approximation in deep learning.**    One component of the SDE approximation is the gradient noise distribution. When the noise is an isotropic Gaussian distribution (i.e., $\Sigma(X_t) \equiv I$), then the equilibrium of the SDE is the Gibbs distribution. Shi et al. (2020) used an isotropic Gaussian noise assumption to derive a convergence rate on SGD that clarifies the role of the LR during training. Several works have relaxed the isotropic assumption but assume the noise is constant. Mandt et al. (2017) assumed the covariance $\Sigma(X)$ is locally constant to show that SGD can be used to perform Bayesian posterior inference. Zhu et al. (2019) argued that when constant but anisotropic SGD noise aligns with the Hessian of the loss, SGD is able to more effectively escape sharp minima. When noise covariance $\Sigma(X_t)$ is uniformly positive definite, Hu et al. (2019) showed that SDE approximation Equation (2) escapes strict saddle points in $O(\ln \eta^{-1})$ time, which matches the $O(\eta^{-1} \ln \eta^{-1})$ escaping rate for SGD (Fang et al., 2019; Jin et al., 2017).

Recently, many works have used the most common form of the SDE approximation (2) with parameter-dependent noise covariance. Li et al. (2020) and Kunin et al. (2020) used the symmetry of loss (scale invariance) to derive properties of dynamics (i.e., $\Sigma(X_t)X_t = 0$). Li et al. (2020) further used this property to explain the phenomenon of sudden rising error after LR decay in training. Smith et al. (2020) used the SDE to derive the linear scaling rule (Goyal et al., 2017) and Definition 2.1 for infinitesimally small LR. Xie et al. (2021) constructed a SDE-motivated diffusion model to propose why SGD favors flat minima during optimization. Cheng et al. (2020) analyzed MCMC-like continuous dynamics and construct an algorithm that provably converges to this limit, although their dynamics do not model SGD.

**Theoretical Foundations of the SDE approximation for SGD.**    Despite the popularity of using SDEs to study SGD, theoretical justification for this approximation has generally relied upon tiny LR (Li et al., 2019a; Hu et al., 2019). Cheng et al. (2020) proved a strong approximation result for an SDE and MCMC-like dynamics, but not SGD. Wu et al. (2020) argued that gradient descent with Gaussian noise can generalize as well as SGD, but their convergence proof also relied on an infinitesimally small LR.

**LR and Batch Size.**    It is well known that using large batch size or small LR will lead to worse generalization (Bengio, 2012; LeCun et al., 2012). According to (Keskar et al., 2017), generalization is harmed by the tendency for large-batch training to converge to sharp minima, but Dinh et al. (2017) argued that the invariance in ReLU networks can permit sharp minima to generalize well too. Li et al. (2019b) argued that the LR can change the order in which patterns are learned in a non-homogeneous

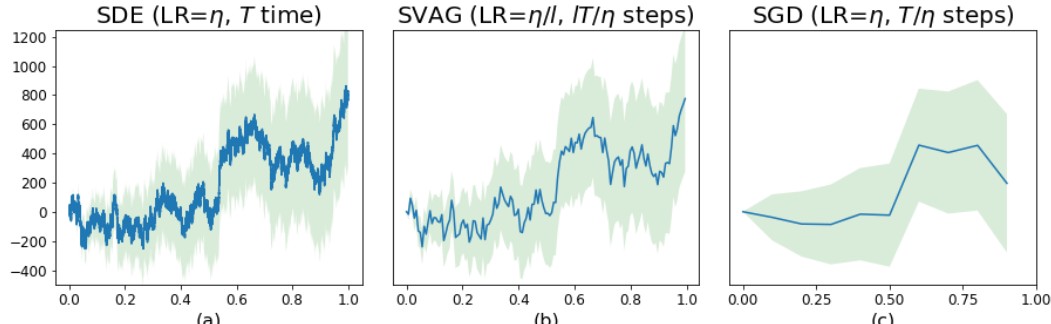

Figure 2: Itô SDE (2), SVAG (5), and SGD (1) trajectories (blue) sampled from a distribution (green). Li et al. (2019a) show that $\forall T, \exists \eta$ such that SDE (a) and SGD (c) are order-1 weak approximations (Definition 4.2) of each other. Our result (Theorem 4.3) shows that $\forall T, \eta, \exists l$ such that SDE (a) and SVAG (b) are order-1 weak approximations of each other. In particular, Li et al. (2019a) requires an infinitesimal $\eta$ and our result holds for finite $\eta$.

synthetic dataset. Several works (Hoffer et al., 2017; Smith and Le, 2018; Chaudhari and Soatto, 2018; Smith et al., 2018) have had success using a larger LR to preserve the scale of the gradient noise and hence maintain the generalization properties of small-batch training. The relationship between LR and generalization remains hazy, as (Shallue et al., 2019) empirically demonstrated that the generalization error can depend on many other training hyperparameters.

# 4 Stochastic Variance Amplified Gradient (SVAG)

Experimental verification of the SDE approximation appears computationally intractable by traditional methods. We provide an algorithm, *Stochastic Variance Amplified Gradient* (SVAG), that efficiently simulates and provably converges to the Itô SDE (2) for a given training setting (Theorem 4.3). Moreover, we use SVAG to experimentally verify that the SDE approximation closely tracks SGD for many common settings (Figure 4; additional settings in Appendix F).

## 4.1 The SVAG Algorithm

For a chosen hyperparameter $l \in \mathbb{N}^+$, we define

$$x_{k+1} = x_k - \frac{\eta}{l} \nabla \mathcal{L}^l_{\bar{\gamma}_k}(x_k), \tag{5}$$

where $\bar{\gamma}_k = (\gamma_{k,1}, \gamma_{k,2})$ with $\gamma_{k,1}, \gamma_{k,2}$ sampled independently and

$$\mathcal{L}^l_{\bar{\gamma}_k}(\cdot) := \frac{1 + \sqrt{2l-1}}{2} \mathcal{L}_{\gamma_{k,1}}(\cdot) + \frac{1 - \sqrt{2l-1}}{2} \mathcal{L}_{\gamma_{k,2}}(\cdot).$$

SVAG is equivalent to performing SGD on a new distribution of loss functions constructed from the original distribution: the new loss function is a linear combination of two independently sampled losses $\mathcal{L}_{\gamma_{k,1}}$ and $\mathcal{L}_{\gamma_{k,2}}$, usually corresponding to the losses on two independent batches. This ensures that the expected gradient is preserved while amplifying the gradient covariance by a factor of $l$, i.e., $\sqrt{\frac{\eta}{l}} \Sigma^l(x) = \sqrt{\eta} \Sigma^1(x)$, where $\Sigma^l(x) := \mathbb{E}[(\nabla \mathcal{L}^l_{\bar{\gamma}}(x) - \nabla \mathcal{L}^l(x))(\mathcal{L}^l_{\bar{\gamma}}(x) - \nabla \mathcal{L}^l(x))^\top]$. Therefore, the Itô SDE that matches the first and second order moments is always (2). We note that SVAG is equivalent to SGD when $l = 1$, and both the expectation and covariance of the one-step update $(x_{k+1} - x_k)$ are proportional to $1/l$, meaning the direction of the update is noisier when $l$ increases.

## 4.2 SVAG Approximates the SDE

**Definition 4.1** (Test Functions). Class $G$ of continuous functions $\mathbb{R}^d \to \mathbb{R}$ has *polynomial growth* if $\forall g \in G$ there exist positive integers $\kappa_1, \kappa_2 > 0$ such that for all $x \in \mathbb{R}^d$, $|g(x)| \leq \kappa_1(1 + |x|^{2\kappa_2})$.

For $\alpha \in \mathbb{N}^+$, we denote by $G^\alpha$ the set of $\alpha$-times continuously differentiable functions $g$ where all partial derivatives of form $\frac{\partial^{\overline{\alpha}} g}{\partial x_1^{\alpha_1} \cdots \partial x_d^{\alpha_d}}$ s.t. $\sum_{i=1}^d \alpha_i = \overline{\alpha} \leq \alpha$, are also in $G$.

**Definition 4.2** (Order-$\alpha$ weak approximation). Let $\{X_t^\eta : t \in [0, T]\}$ and $\{x_k^\eta\}_{k=0}^{\lfloor \frac{T}{\eta} \rfloor}$ be families of continuous and discrete stochastic processes parametrized by $\eta$. We say $\{X_t^\eta\}$ and $\{x_k^\eta\}$ are order-$\alpha$

weak approximations of each other if for every $g \in G^{2(\alpha+1)}$, there is a constant $C > 0$ independent of $l$ such that

$$\max_{k=0,\ldots,\lfloor \frac{T}{\eta} \rfloor} \left| \mathbb{E}g(x_k^\eta) - \mathbb{E}g(X_{k\eta}^\eta) \right| \leq C\eta^\alpha.$$

When applicable, we drop the superscript $\eta$, and say $\{X_t\}$ and $\{x_k\}$ are *order-$\alpha$ (or $\alpha$ order) approximations* of each other.

We now show that SVAG converges weakly to the Itô SDE approximation in (2) when $l \to \infty$, i.e., $x_{lk}$ and $X_{k\eta}$ have the roughly same distribution. Figure 2 highlights the differences between our result and (Li et al., 2019a). Figure 4 provide verification of the below theorem, and additional settings are studied in Appendix F.

**Theorem 4.3.** *Suppose the following conditions[2] are met:*

  *(i)* $\mathcal{L} \equiv \mathbb{E}\mathcal{L}_\gamma$ *is $\mathcal{C}^\infty$-smooth, and $\mathcal{L} \in G^4$.*

  *(ii)* $|\nabla\mathcal{L}_\gamma(x) - \nabla\mathcal{L}_\gamma(y)| \leq L_\gamma|x - y|$, *for all $x, y \in \mathbb{R}^d$, where $L_\gamma > 0$ is a random variable with finite moments, i.e., $\mathbb{E}L_\gamma^k$ is bounded for $k \in \mathbb{N}^+$.*

  *(iii)* $\Sigma^{\frac{1}{2}}(X)$ *is $\mathcal{C}^\infty$-smooth in $X$.*

*Let $T > 0$ be a constant and $l$ be the SVAG hyperparameter (5). Define $\{X_t : t \in [0, T]\}$ as the stochastic process (independent of $\eta$) satisfying the Itô SDE (2) and $\{x_k^{\eta/l} : 1 \leq k \leq \lfloor lT/\eta \rfloor\}$ as the trajectory of SVAG (5) where $x_0 = X_0$. Then, SVAG $\{x_k^{\eta/l}\}$ is an order-1 weak approximation of the SDE $\{X_t\}$, i.e. for each $g \in G^4$, there exists a constant $C > 0$ independent of $l$ such that*

$$\max_{k=0,\ldots,\lfloor lT/\eta \rfloor} |\mathbb{E}g(x_k^{\eta/l}) - \mathbb{E}g(X_{\frac{k\eta}{l}})| \leq Cl^{-1}.$$

**Remark 4.4.** *Lipschitz conditions like (ii) are often not met by deep learning objectives. For instance using normalization schemes can make derivatives unbounded, but if the trajectory $\{x_t\}$ stays bounded away from the origin and infinity, then (ii) holds.*

**Remark 4.5.** *Though technically the weak approximation result (Theorem 4.3) only applies when the stochastic gradient is sampled independently at each step, experimentally we found the difference between performance of SGD and SVAG is negligible among different sampling methods, including random shuffling, sampling with and without replacement. (See detailed discussions in Appendix F.1 and Figure 7) This experimental evidence suggests the validity of SDE approximation doesn't change with sampling methods used in practice.*

### 4.3 Proof Overview

Let $\{X_t^{x,s} : t \geq s\}$ denote the stochastic process obeying the Itô SDE (2) starting from time $s$ and with the initial condition $X_s^{x,s} = x$ and $\{x_k^{x,j} : k \geq j\}$ denote the stochastic process (depending on $l$) satisfying SVAG (5) with initial condition $x_j^{x,j} = x$. For convenience, we define $\widetilde{X}_k := X_{\frac{k\eta}{l}}$ and write $\widetilde{X}_k^{x,j} := X_{\frac{k\eta}{l}}^{x,\frac{j\eta}{l}}$. Alternatively, we write $\widetilde{X}_k(x, j) := \widetilde{X}_k^{x,j}$ and $x_k(x, j) := x_k^{x,j}$.

Now for any $1 \leq k \leq \lfloor \frac{lT}{\eta} \rfloor$, we interpolate between a SVAG solution $x_k$ and SDE solution $\widetilde{X}_k$ through a series of hybrid trajectories $\widetilde{X}_k(x_j, j)$, i.e., the weight achieved by running SVAG for the first $j$ steps and then SDE from time $j$ to $k$. The two limits of the interpolation are $\widetilde{X}_k(x_k, k) = x_k$ (i.e., SVAG solution after $k$ steps) and $\widetilde{X}_k(x_0, 0) = \widetilde{X}_k$ (i.e., SDE solution after $k$ time). This yields the following error decomposition for a test function $g \in G$ (see Definition 4.1).

$$|\mathbb{E}g(x_k) - \mathbb{E}g(X_{\frac{k\eta}{l}})| = |\mathbb{E}g(x_k) - \mathbb{E}g(\widetilde{X}_k)| \leq \sum_{j=0}^{k-1} \left| \mathbb{E}g(\widetilde{X}_k(x_{j+1}, j+1)) - \mathbb{E}g(\widetilde{X}_k(x_j, j)) \right|$$

Note that each pair of adjacent hybrid trajectories only differ by a single step of SVAG or SDE. We show that the one-step increments of SVAG and SDE are close in distribution along the entire trajectory by computing their moments (Lemmas 4.6 and 4.7). Then, using the Taylor expansion of $g$, we can show that the single-step approximation error from switching from SVAG to SDE is uniformly upper bounded by $O(\frac{\eta^2}{l^2})$. (See Figure 6 for demonstration) Hence, the total error is $O(k\frac{\eta^2}{l^2}) = O(\frac{\eta}{l})$.

---

[2]The $\mathcal{C}^\infty$ smoothness assumptions can be relaxed by using the mollification technique in Li et al. (2019a).

**Lemma 4.6.** *Define the one-step increment of the Itô SDE as $\widetilde{\Delta}(x) = \widetilde{X}_1^{x,0} - x$. Then we have*

(i) $\mathbb{E}\widetilde{\Delta}(x) = -\frac{\eta}{l}\nabla\mathcal{L}(x) + \mathcal{O}(l^{-2})$,      (ii) $\mathbb{E}\widetilde{\Delta}(x)\widetilde{\Delta}(x)^\top = \frac{\eta^2}{l}\Sigma(x) + \mathcal{O}(l^{-2})$,

(ii) $\mathbb{E}\widetilde{\Delta}(x)^{\otimes 3} = \mathcal{O}(l^{-2})$,      (iv) $\sqrt{\mathbb{E}|\widetilde{\Delta}(x)^{\otimes 4}|^2} = \mathcal{O}(l^{-2})$.

**Lemma 4.7.** *Define the one-step increment of SVAG as $\Delta(x) = x_1^{x,0} - x$. Then we have*

(i) $\mathbb{E}\Delta(x) = -\frac{\eta}{l}\nabla\mathcal{L}(x)$,

(ii) $\mathbb{E}\Delta(x)\Delta(x)^\top = \frac{\eta^2}{l}\Sigma(x) + \frac{\eta^2}{l^2}\nabla\mathcal{L}(x)\nabla\mathcal{L}(x)^\top = \frac{\eta^2}{l}\Sigma(x) + \mathcal{O}(l^{-2})$,

(iii) $\mathbb{E}\Delta(x)^{\otimes 3} = \frac{\eta^3}{l^2}\frac{3-l^{-1}}{2}\Lambda(x) + \frac{\eta^3}{l^3}\left(3\overline{\nabla\mathcal{L}(x)\otimes\Sigma(x)} + \nabla\mathcal{L}(x)^{\otimes 3}\right) = \mathcal{O}(l^{-2})$

(iv) $\sqrt{\mathbb{E}|\Delta(x)^{\otimes 4}|^2} = \mathcal{O}(l^{-2})$,

*where $\Lambda(x) := \mathbb{E}(\nabla\mathcal{L}_{\gamma_1}(x) - \nabla\mathcal{L}(x))^{\otimes 3}$, and $\overline{\mathcal{T}}$ denotes the symmetrization of tensor $\mathcal{T}$, i.e., $\overline{\mathcal{T}}_{ijk} = \frac{1}{6}\sum_{i',j',k'}\mathcal{T}_{i'j'k'}$, where $i', j', k'$ sums over all permutation of $i, j, k$.*

Though (i) and (ii) in Lemma 4.7 hold for any discrete update with LR $= \frac{\eta}{l}$ that matches the first and second order moments of SDE (2), (iii) and (iv) could fail. For example, when decreasing LR according to LSR (Definition 2.1), even if we can use a fractional batch size and sample an infinitely divisible noise distribution, we may arrive at a different continuous limit if (iii) and (iv) are not satisfied. (See a more detailed discussion in Appendix B.2) SVAG is not the unique way to ensure (iii) and (iv), and any other design (e.g. using three copies per step and with different weights) satisfying Lemma 4.7 are also first order approximations of SDE (2), by the same proof.

## 5   Understanding the Failure of SDE Approximation and LSR

In this section, we analyze how *discretization error*, caused by large LR, leads to the failure of the SDE approximation (Section 5.1) and LSR (Section 5.2) for *scale invariant* networks (e.g., nets equipped with BatchNorm (Ioffe and Szegedy, 2015) and GroupNorm (Wu and He, 2018)). To get best generalization, practitioners often add Weight Decay (WD, a.k.a $\ell_2$ regularization; see (7)). Intriguingly, unlike the traditional setting where $\ell_2$ regularization controls the capacity of function space, for scale invariant networks, each norm ball has the same expressiveness regardless of the radius, and thus WD only regularize the model implicitly via affecting the dynamics. Li et al. (2020) explained such phenomena by showing for training with Normalization, WD and constant LR, the parameter norm converges and WD affects 'effective LR' by controlling the limiting value of the parameter norm. That paper also gave experiments showing that the training loss will reach some plateau, and gave evidence of training reaching an "equilibrium" distribution that it does not get out of unless if some hyperparameter is changed. Throughout this section we assume the existence of equilibrium for SGD and SDE.

To quantify differences in training algorithms, we would ideally work with statistics like the train/test loss and accuracy achieved, but characterizing optimization and generalization properties of deep networks beyond the NTK regime (Jacot et al., 2018; Allen-Zhu et al., 2019b; Du et al., 2019; Arora et al., 2019a; Allen-Zhu et al., 2019a) is in general an open problem. Therefore, we rely on other natural test functions (Definition 5.1).

### 5.1   Failure of SDE Approximation

In Theorem 5.2, we show that the SDE-approximation of SGD is bound to fail for these scale-invariant nets when LR gets too large. Specifically, using above-mentioned results we show that the equilibrium distributions of SGD and SDE are quite far from each other with respect to expectations of these natural test functions (Definition 5.1).

We consider the below SDE (6) with arbitrary expected loss $\mathcal{L}(x)$ and covariance $\overline{\Sigma}(x)$, and the moment-matching SGD (7) satisfying $\mathbb{E}\mathcal{L}_\gamma(x) = \mathcal{L}(x)$ and $\overline{\Sigma}(x) = \eta\Sigma(x)$ where $\Sigma(x)$ is the covariance of $\nabla\mathcal{L}_\gamma(x)$. In the entire Section 5, we will assume that for all $\gamma$, $\mathcal{L}_\gamma$ is *scale invariant* (Arora et al., 2019b; Li and Arora, 2020a), i.e., $\mathcal{L}_\gamma(x) = \mathcal{L}_\gamma(cx), \forall c > 0$ and $x \in \mathbb{R}^d \setminus \{0\}$. [3]

---

[3]The results in this section can be extended straightforwardly to the case where the network is not entirely scale invariant, where the $C$-closeness is defined for the corresponding metrics of the scale-invariant parameters.

$$\mathrm{d}X_t = -\nabla\big(\mathcal{L}(X_t) + \frac{\lambda}{2}|X_t|^2\big)\mathrm{d}t + \overline{\Sigma}^{1/2}(X_t)\mathrm{d}W_t \tag{6}$$

$$x_{k+1} = x_k - \eta\nabla\big(\mathcal{L}_{\gamma_k}(x_k) + \frac{\lambda}{2}|x_k|^2\big) \tag{7}$$

We will measure the closeness of two distributions by three test functions: squared weight norm $|x|^2$, squared gradient norm $|\nabla\mathcal{L}(x)|^2$, and trace of noise covariance $\mathrm{Tr}[\Sigma(x)]$. We say two equilibrium distributions are $C$-*close* if expectations of these test functions are within a multiplicative constant $C$.

**Definition 5.1** ($C$-closeness). Assuming the existence of the following limits, we use

$R_\infty := \lim\limits_{t\to\infty} \mathbb{E}|x_t|^2,$ $\qquad \overline{R}_\infty := \lim\limits_{t\to\infty} \mathbb{E}|X_t|^2,$

$G_\infty := \lim\limits_{t\to\infty} \mathbb{E}|\nabla\mathcal{L}(x_t)|^2,$ $\qquad \overline{G}_\infty := \lim\limits_{t\to\infty} \mathbb{E}|\nabla\mathcal{L}(X_t)|^2,$

$N_\infty := \lim\limits_{t\to\infty} \mathbb{E}[\mathrm{Tr}[\Sigma(x_t)]],$ $\qquad \overline{N}_\infty := \lim\limits_{t\to\infty} \mathbb{E}[\mathrm{Tr}[\overline{\Sigma}(X_t)]]$

to denote the limiting squared weight norm, gradient norm and trace of covariance for SGD (7) and SDE (6). We say the two equilibriums are $C$-*close* to each other iff

$$\frac{1}{C} \le \frac{R_\infty}{\overline{R}_\infty}, \frac{G_\infty}{\overline{G}_\infty}, \frac{\eta N_\infty}{\overline{N}_\infty} \le C. \tag{8}$$

We call $\frac{N_\infty}{G_\infty}$ and $\frac{\overline{N}_\infty}{\overline{G}_\infty}$ the *noise-to-signal ratio (NSR)*, and below we show that it plays an important role. When the LR of SGD significantly exceeds the NSR of the corresponding SDE, the approximation fails. Of course, we lack a practical way to calculate NSR of the SDE so this result is existential rather than effective. Therefore we give a condition in terms of NSR of the SGD that *suffices* to imply failure of the approximation. Experiments later in the paper show this condition is effective at showing divergence from SDE behavior.

**Theorem 5.2.** *If either (i).* $\eta > \frac{\overline{N}_\infty}{\overline{G}_\infty}(C^2 - 1)$ *or (ii).* $\frac{N_\infty}{G_\infty} < \frac{1}{C^2-1}$, *then the equilibria of SDE (6) and SGD (7) are not $C$-close.*

The high-level idea behind Theorem 5.2 is the observation that the norm dynamics of SGD (10) and SDE (9) differ by a second order discretization error related to the norm of full-batch gradient, $\eta^2\mathbb{E}|\nabla\mathcal{L}(x_k)|^2$. Thus intuitively, the two dynamics can be close only when the difference is tiny and negligible. We can make the argument formal by comparing the relationships between the above defined three metrics at the equilibrium of SGD Equation (11) and SDE Equation (12), and thus conclude that a sufficiently large noise-to-signal ratio (NSR) is a necessary condition for $C$-closeness.

Here the role of scale invariance is to simplify the norm dynamics of both SGD (10) and SDE (9) by removing the cross term. This is because of a well-known property of scale-invariance, the orthogonality between gradient and the weight itself, i.e., $\langle\nabla\mathcal{L}_\gamma(x), x\rangle = 0$ for any $x, \gamma$. (Lemma E.7)

*Proof of Theorem 5.2.* We will prove the the contrapositive statement: if the equilibriums of (7) and (6) are $C$-close, then $\eta \le \frac{\overline{N}_\infty}{\overline{G}_\infty}(C^2 - 1)$ and $\frac{1}{C^2-1} \le \frac{N_\infty}{G_\infty}$. Following the derivation in (Li et al., 2020), by Itô's lemma:

$$\frac{\mathrm{d}}{\mathrm{d}t}\mathbb{E}|X_t|^2 = -2\lambda\mathbb{E}|X_t|^2 + \mathbb{E}\,\mathrm{Tr}[\Sigma(X_t)]. \tag{9}$$

Again by the orthogonality between gradient and weight, it can be shown that for SGD (7),

$$\mathbb{E}|x_{k+1}|^2 - \mathbb{E}|x_k|^2 = (1 - \eta\lambda)^2\mathbb{E}|x_k|^2 + \eta^2\mathbb{E}|\nabla\mathcal{L}_{\gamma_k}(x_k)|^2 - \mathbb{E}|x_k|^2$$
$$= \eta\lambda(-2+\eta\lambda)\mathbb{E}|x_k|^2 + \eta^2\mathbb{E}|\nabla\mathcal{L}(x_k)|^2 + \eta^2\mathbb{E}\,\mathrm{Tr}[\Sigma(x_k)] \tag{10}$$

If both $x_k$ and $X_t$ have reached their equilibriums, both LHS of (9) and (10) are 0, and therefore

$$(2 - \eta\lambda)\lambda R_\infty = \eta G_\infty + \eta N_\infty, \tag{11}$$

$$2\lambda\overline{R}_\infty = \overline{N}_\infty. \tag{12}$$

Combining (11), (12), and (8), we have

$$\eta G_\infty + \eta N_\infty \le 2\lambda R_\infty \le 2\lambda C\overline{R}_\infty = C\overline{N}_\infty.$$

Applying (8) again, we have $\eta\overline{G}_\infty + \overline{N}_\infty \le C\eta(G_\infty + N_\infty) \le C^2\overline{N}_\infty \le C^3\eta N_\infty$. The proof is completed by comparing the first and third, the second and the fourth terms respectively. $\qquad\square$

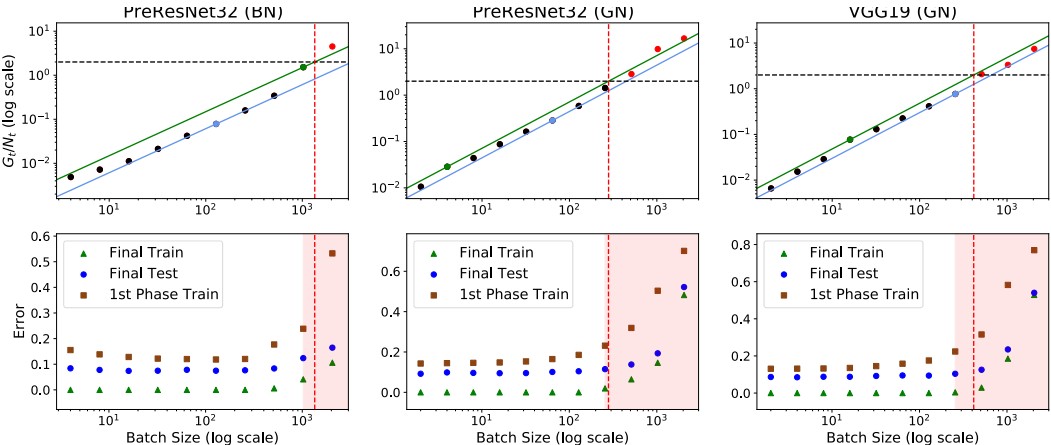

Figure 3: Experimental verification for our theory on predicting the failure of Linear Scaling Rule. We modify PreResNet-32 and VGG-19 to be scale-invariant (according to Appendix C of (Li et al., 2020)). All three settings use the same LR schedule, LR= 0.8 initially and is decayed by 0.1 at epoch 250 with 300 epochs total budget. Here, $G_t$ and $N_t$ are the empirical estimations of $G_\infty$ and $N_\infty$ taken after reaching equilibrium in the first phase (before LR decay). Per the approximated version of Theorem 5.6, i.e., $B^* = \kappa B \lesssim C^2 B N_\infty^B / G_\infty^B$, we use baseline runs with different batch sizes $B$ to report the maximal and minimal predicted critical batch size, defined as the intersection of the threshold ($G_t/N_t = C^2$) with the green and blue lines, respectively. We choose a threshold of $C^2 = 2$, and consider LSR to fail if the final test error exceeds the lowest achieved test error by more than 20% of its value, marked by the red region on the plot. Further settings and discussion are in Appendix F.

**Remark 5.3.** *Since the order-1 approximation fails for large LR, it's natural to ask if higher-order SDE approximation works. In Theorem E.4 we give a partial answer, that the same gap happens already between order-1 and order-2 SDE approximation, when $\eta \gtrsim \frac{\overline{N_\infty}}{\overline{G_\infty}}(C^2 - 1)$. This suggests failure of SDE approximation may be due to missing some second order term, and thus higher-order approximation in principle could avoid such failure. On the other hand, when approximation fails in such ways, e.g., increasing batch size along LSR, the performance of SGD degrades while SDE remains good. This suggests the higher-order correction term may not be very helpful for generalization.*

## 5.2  Failure of Linear Scaling Rule

In this section we derive a similar necessary condition for LSR to hold.

Similar to Definition 5.1, we will use $R_\infty^{B,\eta}, G_\infty^{B,\eta}, N_\infty^{B,\eta}$ as test functions for equilibrium achieved by SGD (7) when training with LR $\eta$ and mini-batches of size $B$. We first introduce the concept of Linear Scaling Invariance (LSI). Note here we care about the scaled ratio $N_\infty^{B,\eta}/(\kappa N_\infty^{\kappa B,\kappa\eta})$ because the covariance scales inversely to batch size, $\Sigma^B(x) = \kappa \Sigma^{\kappa B}(x)$.

**Definition 5.4** ($(C,\kappa)$-Linear Scaling Invariance). We say SGD (7) with batch size $B$ and LR $\eta$ exhibits $(C,\kappa)$-LSI if, for a constant $C$ such that $0 < C < \sqrt{\kappa}$,

$$\frac{1}{C} \leq \frac{R_\infty^{B,\eta}}{R_\infty^{\kappa B,\kappa\eta}}, \frac{N_\infty^{B,\eta}}{\kappa N_\infty^{\kappa B,\kappa\eta}}, \frac{G_\infty^{B,\eta}}{G_\infty^{\kappa B,\kappa\eta}} \leq C. \tag{13}$$

We show below that $(C,\kappa)$-LSI fails if the NSR $\frac{N_\infty}{G_\infty}$ is too small, thereby giving a certificate for failure of $(C,\kappa)$-LSI even without a baseline run.

**Theorem 5.5.** *For any $B$, $\eta$, $C$, and $\kappa$ such that*

$$\frac{N_\infty^{\kappa B,\kappa\eta}}{G_\infty^{\kappa B,\kappa\eta}} < (1 - \frac{1}{\kappa})\frac{1}{C^2 - 1} - \frac{1}{\kappa}, \tag{14}$$

*SGD with batch size $B$ and LR $\eta$ does not exhibit $(C,\kappa)$-LSI.*

We now present a simple and efficient procedure to find the largest $\kappa$ for which $(C,\kappa)$-LSI will hold, providing useful guidance to make hyper-parameter tuning more efficient. Before doing so, one must

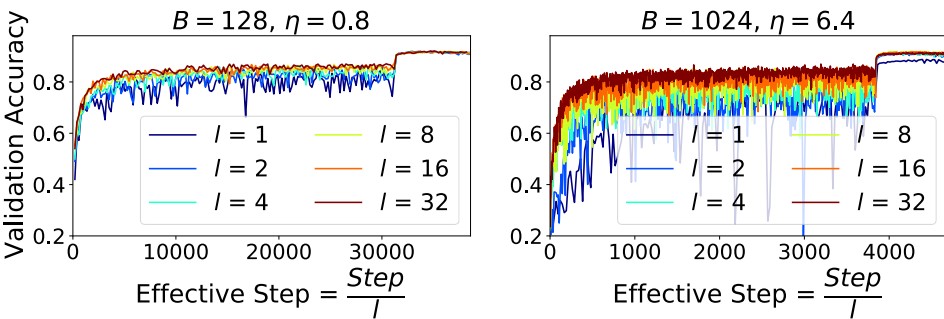

Figure 4: SVAG converges quickly and matches SGD (left) or shows the failure of the SDE approximation when LSR breaks (right). We train PreResNet32 with BN on CIFAR-10 for 300 epochs, decaying $\eta$ by 0.1 at epoch 250. SVAG takes $l$ smaller steps to simulate the continuous dynamics in $\eta$ time, so we plot the accuracy against "effective steps," and we note that SVAG with $l = 1$ is equivalent to SGD. We predict in Figure 3 that LSR (and thus, the SDE approximation) breaks at $B = 1024$ for this training setting, and here we observe SVAG converges to a limiting trajectory different from SGD, suggesting that the SDE approximation did indeed break.

choose an appropriate value for $C$, which controls how close the test functions must be for us to consider LSR to have "worked." It is an open question what value of $C$ will ensure that the two settings achieve similar test performance, but throughout our experiments across various datasets and architectures in Figure 3 and Appendix F, we find that $C = \sqrt{2}$ works well. One can estimate $G_\infty^{B,\eta}$ and $N_\infty^{B,\eta}$ from a baseline run. Then, one can straightforwardly compute the value for the $\kappa$ threshold given in the theorem below. We conduct this process in Figure 3 and Appendix F to test our theory.

**Theorem 5.6.** *For any $B$, $\eta$, $C$, and*

$$\kappa > C^2\left(1 + \frac{N_\infty^{B,\eta}}{G_\infty^{B,\eta}}\right), \quad \left(\approx C^2 \frac{N_\infty^{B,\eta}}{G_\infty^{B,\eta}} \text{ when } \frac{N_\infty^{B,\eta}}{G_\infty^{B,\eta}} \gg 1\right), \tag{15}$$

*SGD with batch size $B$ and LR $\eta$ does not exhibit $(C, \kappa)$-LSI.*

## 6 Experiments

We provide our code at `https://github.com/sadhikamalladi/svag`. Figure 3 provides experimental evidence that measurements from a single baseline run can be used to predict when LSR will break, thereby providing verification for Theorem 5.6. Surprisingly, it turns out the condition in Theorem 5.6 is not only sufficient but also close to necessary. Figure 4 and Appendix F.1 test SVAG on common architectures and datasets and report the results. Theorem 4.3 shows that SVAG converges to the SDE as $l \to \infty$, but we note that SVAG needs $l$ times as many steps as SGD to match the SDE. Therefore, in order for SVAG to be a computationally efficient simulation of the SDE, we hope to observe convergence for small values of $l$. This is confirmed in Figure 4 and Appendix F.1. The success of SVAG in matching SGD in many cases indicates that studying the Itô SDE can yield insights about the behavior of SGD. We note our experiments are limited in the sense that it only confirms the closeness of train/test accuracy curves, which doesn't verify the weak convergence guaranteed in Theorem 4.3. But at least, the experiments suggest that SDE and SVAG with large $l$ are interesting learning algorithms to study, with similar or even better generalization than SGD. Moreover, in the case where we expect the SDE approximation to fail (e.g., when LSR fails), SVAG does indeed converge to a different limiting trajectory from the SGD trajectory.

## 7 Conclusion

We present a computationally efficient simulation SVAG (Section 4) that provably converges to the canonical order-1 SDE (2), which we use to verify that the SDE is a meaningful approximation for SGD in common deep learning settings (Section 6). We relate the discretization error to LSR (Definition 2.1): in Section 5 we derive a testable necessary condition for the SDE approximation and LSR to hold, and in Figure 3 we demonstrate its applicability to standard settings.

## Acknowledgement

The authors acknowledge support from NSF, ONR, Simons Foundation, DARPA and SRC. ZL is also supported by Microsoft Research PhD Fellowship.

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
