# A   Preliminaries on SDE

## A.1   SDE Approximation Schemes

Here, we review the common approximation schemes for SDEs and discuss why they are not efficient enough to be applied to the Itô SDE approximation for SGD. We adapt the information in Chapters 13 and 14 of Kloeden and Platen (2011). In general, an Itô SDE can be written as

$$dX_t = \mu(X_t, t)dt + \sigma(X_t, t)dW_t$$

where $\mu$ and $\sigma$ are called the drift and diffusion coefficients respectively. The standard Itô SDE (2) used to approximate SGD sets $\mu(X_t, t) = -\nabla\mathcal{L}(X_t)$ and $\sigma(X_t, t) = (\eta\Sigma(X_t))^{1/2}$.

Suppose we want to solve the SDE on a time interval $[0, T]$. First, we discretize the time interval into $N$ equal steps $\tau_1, ..., \tau_N$ of size $\Delta t$. We will construct a Markov chain $Y$ that is a weak approximation in $\Delta t$ (Definition 4.2) to the true solution, and let $Y_0 = x_0$ where $x_0$ is the initialization for the SGD trajectory.

The *Euler-Maruyama* scheme is the simplest approximation scheme, and the resulting Markov chain is an order 1 weak approximation to the true solution of the SDE. For $n \in \mathbb{N}$, $0 \leq n \leq N - 1$,

$$Y_{n+1} = Y_n + \mu(Y_n, \tau_n)\Delta t + \sigma(Y_n, \tau_n)\Delta W_n \tag{16}$$

where $\Delta W_n \overset{i.i.d.}{\sim} \mathcal{N}(0, \Delta t)$. In the ML setting, computing a single step in this Markov chain requires computing the full gradient (for $\mu(Y_n, \tau_n)$) and the covariance of the gradient (for $\sigma(Y_n, \tau_n)$). As such, modeling a single step in the recurrence requires making one pass over the entire dataset. The error of the approximation scheme scales with $\Delta t$, so making $N$ larger (thereby requiring more recurrence steps) will improve the quality of the approximate solution. We furthermore note that storing the gradient covariance matrix requires a large amount of memory. Each weight parameter in the network must be modeled by its own recurrence equation, so for modern day deep networks, this approximation seems computationally intractable.

The Euler-Maruyama scheme is considered the simplest approximation scheme for an Itô SDE. A variety of other schemes, such as the Milstein and stochastic Runge-Kutta schemes, have been derived by adding a higher order corrective term, taken from the stochastic Taylor expansion, to the recurrence computation. In particular, these schemes all still require the computation of $\mu$ and $\sigma$ at each step of the recurrence, so they remain computationally intractable for the Itô SDE used to approximate SGD.

## A.2   Preliminary on Stochastic Process

**Definition A.1.** We call a $m$-dimensional stochastic process $X = \{X_t : t \geq 0\}$ a Lévy process if it satisfies the following properties:

- $X_0 = 0$ almost surely;
- Independence of increments: For any $0 \leq t_1 < t_2 < \cdots < t_n < \infty$, $X_{t_2} - X_{t_1}, X_{t_3} - X_{t_2}, \ldots, X_{t_n} - X_{t_{n-1}}$ are independent;
- Stationary increments: For any $s < t$, $X_t - X_s$ is equal in distribution to $X_{t-s}$;
- Continuity in probability: For any $\varepsilon > 0$ and $t \geq 0$ it holds that $\lim_{h \to 0} P(|X_{t+h} - X_t| > \varepsilon) = 0$.

**Definition A.2.** We call a counting process $\{N(t) : t \geq 0\}$ a Poisson process with rate $\lambda > 0$ if it satisfies the following properties:

- $N(0) = 0$;
- has independence of increments;
- the number of events (or points) in any interval of length $t$ is a Poisson random variable with parameter (or mean) $\lambda t$.

# B   Discussion on Non-Gaussian Noise

In Appendix B.1 we give an example where LSR holds while SDE approximation breaks. In Appendix B.2, we show this example to a more general setting – infinitely divisible noise. We also explain why decreasing LR along LSR will not get a better approximation for SDE, while decreasing LR along SVAG will, since both operation preserves the same SDE approximation. In Appendix B.3, we discuss the possibility where the noise is heavy-tailed and with unbounded covariance.

## B.1 LSR can hold when SDE approximation breaks

**Example B.1.** Let $Z(t)$ be a 1-dimensional Poisson process (Definition Definition A.2), where $Z(t)$ follows Poisson distribution with parameter $t$. We assume the distribution of the gradient on single sampled data $\gamma$, $\nabla L_\gamma(x)$ is the same as $Z(1)$ for any parameter $x$. For a batch $\boldsymbol{B}$ of size B (with replacement), since Poisson process has independent increments, $\nabla \mathcal{L}_{\boldsymbol{B}}(x) := \frac{1}{B}\sum_{\gamma \in \boldsymbol{B}} \nabla \mathcal{L}_\gamma(x) \stackrel{d}{=} \frac{Z(B)}{B}$.

Thus for any constant $T$ and initialization $x_0 = 0$, performing SGD starting from $x_0$ for $\frac{T}{B}$ steps with LR $B\eta$ and batch size $B$, the distribution of $x_{\frac{T}{B}}$ is independent of $B$, i.e.,

$$x_k = x_{k-1} - B\eta \nabla \mathcal{L}_{\boldsymbol{B}_k}(x_{k-1}) \implies x_{\frac{T}{B}} \stackrel{d}{=} -\eta \underbrace{(Z(B) + Z(B) + \cdots + Z(B))}_{\frac{T}{B}\text{'s } Z(B)} \stackrel{d}{=} -\eta Z(T).$$

Thus LSR holds for all batch size $B$. Below we consider the corresponding NGD (3), $\{\hat{x}_k\}$, where

$$\begin{aligned}
\hat{x}_k =& \hat{x}_{k-1} - B\eta \mathbb{E}\nabla \mathcal{L}_{\boldsymbol{B}_k}(\hat{x}_{k-1}) + B\eta\sqrt{\mathbb{E}(\nabla \mathcal{L}_{\boldsymbol{B}_k} - \mathbb{E}\nabla \mathcal{L}_{\boldsymbol{B}_k})^2} z_{k-1} \\
=& \hat{x}_{k-1} - B\eta \mathbb{E}\frac{Z(B)}{B} + B\eta\sqrt{\mathbb{E}(\frac{Z(B)}{B} - \mathbb{E}\frac{Z(B)}{B})^2} z_{k-1} z_{k-1} \\
=& \hat{x}_{k-1} - B\eta + \eta\sqrt{B\mathbb{E}(Z(1) - \mathbb{E}Z(1))^2} z_{k-1} \\
=& \hat{x}_{k-1} - B\eta + \eta\sqrt{B} z_{k-1},
\end{aligned}$$

and $\{z_i\}_{i=0}^{\frac{T}{B}-1} \stackrel{i.i.d.}{\sim} N(0,1)$.

Thus it holds that $\hat{x}_{\frac{T}{B}} = -\eta T + \eta \sum_{k=0}^{\frac{T}{B}-1} z_{k-1} \stackrel{d}{=} -\eta(T + W_T)$, where $W_T$ is a Wiener process with $W_0 = 0$, meaning the NGD final iterate is also independent of $B$, and constant away form the final iterate $x_{\frac{T}{B}} \stackrel{d}{=} -\eta Z(T)$. Indeed we can show the same result for Itô SDE (2), $dX_t = -dt + \sqrt{\eta}dW_t$:

$$X_{\frac{T}{B}\cdot B\eta} = X_{\eta T} = \int_{t=0}^{\eta T} -dt + \sqrt{\eta}dW_t = -\eta T + \sqrt{\eta}W_{\eta T} \stackrel{d}{=} -\eta T + \eta W_T.$$

Thus we conclude that **LSR holds but SDE approximation fails**. Since NGD achieves the same distribution as Itô SDE, **the gap is solely caused by non-gaussian noise**.

However, the reader might still wonder, since batch size is always at least 1, there's always a lower bound for LR $\eta$ when going down along the ladder of LSR, and thus a discrete process with a finite step size of course cannot be approximated by a continuous one arbitrarily well. So isn't this example trivial? In Appendix B.2, we will see even if we are allowed to use fractional batch size, and thus allow $\eta \to 0$, LSR can still hold without Itô SDE approximation.

## B.2 Infinitely Divisible Noise and Lévy SDE

To understand why decreasing LR along LSR will not get a better approximation for SDE, and how LSR can hold without Itô SDE approximation when $\eta \to 0$, we assume the noise is infinitely divisible below for simplicity, which allows us to define SGD with fractional batch sizes and thus we can take the limit of $\eta \to 0$ along the ladder of LSR.

That is, for the original stochastic loss $\nabla \mathcal{L}_\gamma$, for any $m \in \mathbb{N}^+$, there is a random loss function $\mathcal{L}_{\gamma'}^m$, such that $\forall x \in \mathbb{R}^d$, the original stochastic gradient $\nabla \mathcal{L}_\gamma(x)$ is equal in distribution to the sum of $m$ i.i.d. copies of $\nabla \mathcal{L}_\gamma^m(x)$:

$$\nabla \mathcal{L}_\gamma(x) \stackrel{d}{=} \sum_{i=1}^{m} \nabla \mathcal{L}_{\gamma_i'}^m(x). \tag{17}$$

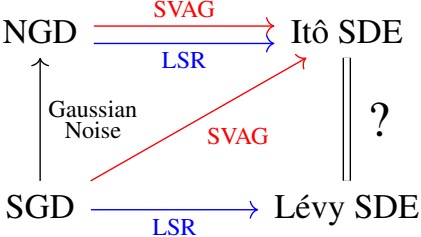

Figure 5: Taking $\eta \to 0$ and keeping the first two moments are not enough to converge to Itô SDE limit, e.g. decreasing LR along LSR can converge to another limit, Lévy SDE. Red and blue arrows means taking limit of the dynamics when LR $\eta \to 0$ along the SVAG and LSR respectively. Here we assume the noise in SGD is infinitely divisible such that the LR can go to 0 along LSR. For NGD, i.e., SGD with Gaussian noise, both SVAG and LSR (Linear Scaling Rule) approaches the same continuous limit. This does not hold for SGD with non-Gaussian noise.

For SGD with batch size $B$, such a random loss function can be found when $m$ is a factor of $B$, where it suffices to define $\mathcal{L}^m$ as $m$ times the same loss with a smaller batch size $\frac{B}{m}$.[4] In other words, we can phrase LSR in a more general form, which only involves the distribution of the noise, but not the generating process of the noise (e.g. noise from sampling a batch with replacement).

**Definition B.2** (generalized Linear Scaling Rule (gLSR)). Keep LR the same. Replace $\nabla\mathcal{L}_\gamma$ by $\nabla\mathcal{L}_\gamma^m$ and multiply the total number of steps by $m$.

It's well known that every infinitely divisible distribution corresponds to a $d$-dimensional Lévy process (Definition A.1) $Z_x(t) \in \mathbb{R}^d$, in the sense that $\nabla\mathcal{L}_\gamma(x) \stackrel{d}{=} Z_x(1)$ Ken-Iti (1999). If we further assume there is a $m$-dimensional Lévy process $Z'(t)$ and a function $\sigma(x) : \mathbb{R}^d \to \mathbb{R}^{d \times m}$ such that for every $x$, $\nabla\mathcal{L}_\gamma(x) - \nabla\mathcal{L}(x) \stackrel{d}{=} \sigma(x)Z(1)$, then by Theorem 2.2 in (Protter et al., 1997), SGD (18) will converge to a limiting continuous dynamic, which we denote as the *Lévy SDE*, as the LR decreases to 0 along LSR.

$$x_k = x_{k-1} - \eta\nabla\mathcal{L}_{\gamma_{k-1}}^m(x_{k-1}), \quad \forall k = 1, \dots, \lfloor\frac{Tm}{\eta}\rfloor \tag{18}$$

Formally, $X_t$ is the solution of the following SDE driven by a Lévy process.

$$\mathrm{d}X_t = -\nabla\mathcal{L}(X_t)\mathrm{d}t + \eta\sigma(X_t)\mathrm{d}Z_{t/\eta}, \quad \forall t \in [0, T] \tag{19}$$

In the special case where $Z_t$ is the $d$-dimensional Brownian motion (note $\{\eta Z_{t/\eta}\}_{t\geq0}$ and $\{\sqrt{\eta}Z_t\}_{t\geq0}$ have the same distributions for the sample paths), $\sigma(x) \in \mathbb{R}^{d \times d}$ will be the square root of the noise covariance, $\Sigma^{\frac{1}{2}}(x)$.

**Why decreasing LR along LSR will not get a better approximation for SDE:** The Lévy SDE is equal to the Itô SDE only when the noise is strictly Gaussian. Thus the gap induced by non-Gaussian noise will not vanish even if both SGD and NGD decrease the LR along LSR, as it will converge to the gap between Itô SDE and Lévy SDE. See Figure 5 for a summary of the relationships among SGD, NGD, Itô SDE, and Lévy SDE.

Since decreasing LR along LSR converges to a different limit than SVAG does, it's natural to ask which part of the approximation in Lemma 4.7 fails for the former. By scrutinizing the proof of Lemma 4.7, we can see (i) and (ii) still hold for any stochastic discrete process with LR $\frac{\eta}{l}$ and matching first and second order moments, while the term $\frac{\eta^3(3-l^{-1})}{2l^2}\Lambda(x)$ now becomes $\frac{\eta^3}{l}\Lambda(x)$ for SGD along LSR, which is larger by an order of $l$. Therefore, the single-step approximation error becomes $O(l^{-1})$ and the total error after $\lfloor Tl/\eta \rfloor$ steps remains constant.[5]

**SDE approximation is not necessary for LSR, even for LR $\eta \to 0$:** We also note that though Smith et al. (2020) derives LSR by assuming the Itô SDE approximation holds, this is only a sufficient

---

[4]Batch loss of nets with BatchNorm is not necessarily divisble, because (17) doesn't hold, as the individual loss depends on the entire batch of data with the presence of BN. Still, it holds for ghost BatchNorm (Hoffer et al., 2017) with $B$ equal to the number of mini-ghost batches.

[5]Such error does not only occur in the third order moment. It also appears in the higher moments. Therefore simply assuming the noise distribution is symmetric (thus $\Lambda = 0$) won't fix this gap.

but not necessary condition for LSR. In Section B.1, we provide a concrete example where LSR holds for all LRs and batch sizes, but the dynamics are constantly away from Itô SDE limit. The loss landscape and noise distribution are constant, i.e., parameter-independent. This is also an example where the gap between SGD and Itô SDE is solely caused by non-gaussian noise, but not the discretization error.

## B.3 Heavy-tailed Noise and Unbounded Covariance

Simsekli et al. (2019) experimentally found that the distribution of the SGD noise appears to be heavy-tailed and proposed to model it with an $\alpha$-stable process. In detail, in Figure 1 of (Simsekli et al., 2019), they show that the histogram of the gradient noise computed with AlexNet on CIFAR-10 is more close to that of $\alpha$-stable random variables, instead of that of Gaussian random variables. However, a more recent paper (Xie et al., 2021) pointed out a fundamental limitation of methodology in (Simsekli et al., 2019): (Simsekli et al., 2019) made a hidden but very restrictive assumption that the noise of each parameter in the model is distributed identically. Moreover, their test (Theorem B.3) of the tail-index $\alpha$ works only under this assumption. Thus the empirical measurement in (Simsekli et al., 2019) ($\widehat{\alpha} < 2$) doesn't exclude the possibility that that stochastic gradient noise follows a joint multivariate Gaussian.

**Theorem B.3.** *(Mohammadi et al., 2015) Let $\{X_i\}_{i=1}^K$ be a collection of i.i.d. random variables with $X_1 \sim \mathcal{S}\alpha\mathcal{S}(\sigma)$ and $K = K_1 \times K_2$. Define $Y_i := \sum_{j=1}^{K_1} X_{j+(i-1)K_1}$ for $i \in \{1, \ldots, K_2\}$. Then the estimator*

$$\widehat{\frac{1}{\alpha}} := \frac{1}{\log K_1} \left( \frac{1}{K_2} \sum_{i=1}^{K_2} \log |Y_i| - \frac{1}{K} \sum_{i=1}^{K} \log |X_i| \right). \tag{20}$$

*converges to $\frac{1}{\alpha}$ almost surely, as $K_2 \to \infty$. Here $\mathcal{S}\alpha\mathcal{S}(\sigma)$ is the $\alpha$-stable distribution defined by $X \sim \mathcal{S}\alpha\mathcal{S}(\sigma) \iff \mathbb{E}[\exp(iwX)] = \exp(-|\sigma w|^\alpha)$.*

We provide the following theoretical and experimental evidence on vision tasks to support the argument in Xie et al. (2021) that it is reasonable to model the stochastic gradient noise by joint Gaussian random variables instead of $\alpha$-stable random variables even *for finite learning rate*. (Note SVAG (e.g., Figure 4) only shows that when LR becomes infinitesimally small, replacing the noise by Gaussian noise gets similar performance.)

1. In Figure 3, we find that the trace of covariance of noise is bounded and the empirical average doesn't grow with the number of samples/batches (this is not plotted in the current paper). However, an $\alpha$-stable random variable has unbounded variance for $\alpha < 2$.
2. In Figures 1, 19, 18, and 20, we show directly that replacing the stochastic gradient noise by Gaussian noise with the same covariance gets almost the train/test curve and the final performance.
3. Applying the test in Theorem B.3 on joint multivariate Gaussian random variables can yield an estimate ranged from 1 to 2 for the tail-index $\alpha$, but for Gaussian variables, $\alpha = 2$. (Theorem B.4)

Another recent work Zhang et al. (2020) also confirmed that the noise in stochastic gradient in ResNet50 on vision tasks is finite. However, they also found the noise for BERT on Wikipedia+Books dataset could be heavy-tailed: the empirical variance is not converging even with $10^7$ samples. We left it as a future work to investigate how does SDE approximate SGD on those tasks or models with heavy-tailed noise.

**Theorem B.4.** *Let $K = K_1 \times K_2 = d \times m \times K_2$, where $K_1, K_2, d, m \in \mathbb{N}^+$. Let $\{X_i\}_{i=1}^K$ be a collection of random variables where $X_{(j-1)d:jd} \overset{i.i.d.}{\sim} N(0, \Sigma)$, for each $j \in \{1, \ldots, mK_2\}$. Then we have*

$$\mathbb{E}\left[ \frac{1}{\log K_1} \left( \frac{1}{K_2} \sum_{i=1}^{K_2} \log |Y_i| - \frac{1}{K} \sum_{i=1}^{K} \log |X_i| \right) \right] = \frac{1}{2} \frac{\log m + \log \mathbf{1}^\top \Sigma \mathbf{1} - \frac{1}{d} \sum_{i=1}^d \log \Sigma_{ii}}{\log m + \log d}.$$

*Specifically, when $d = K_1$ and $m = 1$, taking $\Sigma = \beta \mathbf{1}\mathbf{1}^\top + (1 - \beta)I$, we have*

$$\mathbb{E}\left[ \frac{1}{\log K_1} \left( \frac{1}{K_2} \sum_{i=1}^{K_2} \log |Y_i| - \frac{1}{K} \sum_{i=1}^{K} \log |X_i| \right) \right] = \frac{1}{2} \frac{\log(\beta d^2 + (1 - \beta)d)}{\log d},$$

*and*

$$\left\{ \frac{1}{2} \frac{\log(\beta d^2 + (1-\beta)d)}{\log d} \mid \beta \in [0,1] \right\} = [\frac{1}{2}, 1].$$

*Proof.*

$$\mathbb{E}\left[ \frac{1}{\log K_1} \left( \frac{1}{K_2} \sum_{i=1}^{K_2} \log |Y_i| - \frac{1}{K} \sum_{i=1}^{K} \log |X_i| \right) \right] = \mathbb{E}\left[ \frac{1}{\log K_1} \left( \log |Y_1| - \frac{1}{d} \sum_{i=1}^{d} \log |X_i| \right) \right].$$
(21)

Note $Y_1$ is gaussian with standard deviation $\sqrt{\mathbf{1}^\top \Sigma \mathbf{1} \cdot m}$ and $X_i$ is gaussian with standard deviation $\sqrt{\Sigma_{ii}}$. Thus $\mathbb{E}[\log |Y_1| - \log |X_i|] = \log \sqrt{\mathbf{1}^\top \Sigma \mathbf{1} \cdot m} - \log \sqrt{\Sigma_{ii}}$. Thus we have

$$\mathbb{E}\left[ \frac{1}{\log K_1} \left( \frac{1}{K_2} \sum_{i=1}^{K_2} \log |Y_i| - \frac{1}{K} \sum_{i=1}^{K} \log |X_i| \right) \right]$$
(22)

$$= \mathbb{E}\left[ \frac{1}{\log K_1} \left( \log |Y_1| - \frac{1}{d} \sum_{i=1}^{d} \log |X_i| \right) \right]$$
(23)

$$= \frac{1}{2} \frac{\log m + \log \mathbf{1}^\top \Sigma \mathbf{1} - \frac{1}{d} \sum_{i=1}^{d} \log \Sigma_{ii}}{\log m + \log d}$$
(24)

$\square$

## C   Omitted Derivation in Section 4

We prove Theorem 4.3 in this section. The derivation is based on the following two-step process, following the agenda of (Li et al., 2019a):

1. Showing that the approximation error on a finite interval ($N = \lfloor \frac{Tl}{\eta} \rfloor$ steps) can be upper bounded by the sum of expected one-step errors. (Theorem C.1, which is Theorem 3 in (Li et al., 2019a))

2. Showing the one-step approximation error of SVAG is of order 2, and so the approximation on a finite interval is of order 1. (Lemmas 4.6 and 4.7)

### C.1   Relating one-step to $N$-step approximations

Let us consider generally the question of the relationship between one-step approximations and approximations on a finite interval. Let $T > 0$ and $N = \lfloor lT/\eta \rfloor$. Let us also denote for convenience $\widetilde{X}_k := X_{\frac{k\eta}{l}}$. Further, let $\{X_t^{x,s} : t \geq s\}$ denote the stochastic process obeying the same Equation (2), but with the initial condition $X_s^{x,s} = x$. We similarly write $\widetilde{X}_k^{x,j} := X_{\frac{k\eta}{l}}^{x, \frac{j\eta}{l}}$ and denote by $\{x_k^{x,j} : k \geq j\}$ the stochastic process (depending on $l$) satisfying Equation (5) but with $x_j = x$.

Now, let us denote the one-step changes

$$\text{SVAG:} \quad \Delta(x) := x_1^{x,0} - x, \qquad \text{SDE:} \quad \widetilde{\Delta}(x) := \widetilde{X}_1^{x,0} - x. \tag{25}$$

The following result is adapted from (Li et al., 2019a) to our setting, which relates one-step approximations with approximations on a finite time interval. To prove it, we will construct hybrid trajectories interpolating between SVAG (5) and the SDE (2), as shown in Figure 6.

**Theorem C.1** (Adaption of Theorem 3 in (Li et al., 2019a))**.** *Suppose the following conditions hold:*

(i) *There is a function $K_1 \in G$ independent of $l$ such that* $\left| \mathbb{E}\Delta(x)^{\otimes s} - \mathbb{E}\widetilde{\Delta}(x)^{\otimes s} \right| \leq K_1(x)l^{-2}$

   *for $s = 1, 2, 3$ and* $\sqrt{\mathbb{E} |\Delta(x)^{\otimes 4}|^2} \leq K_1(x)l^{-2}$.

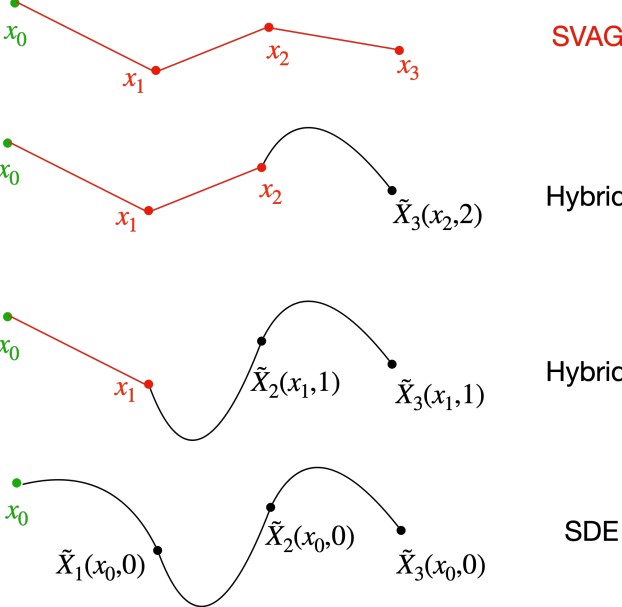

Figure 6: To relate the one-step error to the error over a finite interval, we construct interpolating hybrid trajectories between SVAG and SDE as shown in the figure. Each hybrid trajectory is built by using the second to last SVAG point in the previous trajectory as the initial condition and then running the SDE for the remainder of the time interval.

(ii) For all $m \geq 1$, the $2m$-moment of $x_k^{x,0}$ is uniformly bounded w.r.t. $k$ and $l$, i.e. there exists a $K_2 \in G$, independent of $l, k$, such that $\mathbb{E}|x_k^{x,0}|^{2m} \leq K_2(x)$, for all $k = 0, \ldots, \lfloor lT/\eta \rfloor$.

Then, for each $g \in G^4$, there exists a constant $C > 0$, independent of $l$, such that

$$\max_{k=0,\ldots,\lfloor lT/\eta \rfloor} \left| \mathbb{E}g(x_k) - \mathbb{E}g(X_{\frac{k\eta}{l}}) \right| \leq Cl^{-2}$$

*Proof of Theorem C.1.* Let $T, l > 0$, $N = \lfloor lT/\eta \rfloor$ and for convenience we also define $\widetilde{X}_k := X_{\frac{k\eta}{l}}$. Further, let $\{X_t^{x,s} : t \geq s\}$ denote the stochastic process obeying the same Equation (2), but with the initial condition $X_s^{x,s} = x$. We similarly write $\widetilde{X}_k^{x,j} := X_k^{x,\frac{j\eta}{l}}$ and denote by $\{x_k^{x,j} : k \geq j\}$ the stochastic process (depending on $l$) satisfying Equation (5) but with $x_j = x$. Alternatively, we write $\widetilde{X}_k(x,j) := \widetilde{X}_k^{x,j}$ and $x_k(x,j) := x_k^{x,j}$. By definition, $\widetilde{X}_k(x_k, k) = x_k$ and $\widetilde{X}_k(x_0, 0) = \widetilde{X}_k$.

Thus we have for any $1 \leq k \leq \lfloor \frac{lT}{\eta} \rfloor$, we can decompose the error as illustrated in Figure 6,

$$|\mathbb{E}g(x_k) - \mathbb{E}g(X_{\frac{k\eta}{l}})| = |\mathbb{E}g(x_k) - \mathbb{E}g(\widetilde{X}_k)|$$

$$\leq \sum_{j=0}^{k-1} \left| \mathbb{E}g(\widetilde{X}_k(x_{j+1}, j+1)) - \mathbb{E}g(\widetilde{X}_k(x_j, j)) \right|$$

$$\leq \sum_{j=0}^{k-1} \left| \mathbb{E}u^{k,j+1}(\widetilde{X}_{j+1}(x_j, j)) - \mathbb{E}u^{k,j+1}(x_{j+1}(x_j, j)) \right|$$

$$\leq \sum_{j=0}^{k-1} \left| \mathbb{E}u^{k,j+1}(\widetilde{X}_1(x_j, 0)) - \mathbb{E}u^{k,j+1}(x_1(x_j, 0)) \right|$$

$$\leq \sum_{j=0}^{k-1} \mathbb{E}\left[ \left| \mathbb{E}u^{k,j+1}(\widetilde{X}_1(x_j, 0)) - \mathbb{E}u^{k,j+1}(x_1(x_j, 0)) \right| \big| x_j \right],$$

where $u^{k,j+1}(x)$ is defined as $\mathbb{E}g(X_k(x, j+1))$ and the second to the last step is because of SDE (2) is time-homogeneous. By Proposition 25 in (Li et al., 2019a), $u^{k,j+1} \in G^4$ uniformly, thus by Lemma C.2, we know there exists $K(x) = \kappa_1(1 + |x|^{2\kappa_2}) \in G$ such that

$$|\mathbb{E}g(x_k) - \mathbb{E}g(X_{\frac{k\eta}{l}})| \leq \sum_{j=0}^{k-1} \mathbb{E}\left[K(x_j)l^{-2}\right] \leq \sum_{j=0}^{k-1} \mathbb{E}\left[\kappa_1(1 + |x_j|^{2\kappa_2})l^{-2}\right]$$

By assumption (ii), we know the there is some $K' \in G$,

$$|\mathbb{E}g(x_k) - \mathbb{E}g(X_{\frac{k\eta}{l}})| \leq \sum_{j=0}^{k-1} \mathbb{E}\left[\kappa_1(1 + |x_j|^{2\kappa_2})l^{-2}\right] \leq \sum_{j=0}^{\lfloor \frac{lT}{\eta} \rfloor - 1} \mathbb{E}\left[\kappa_1(1 + |x_j|^{2\kappa_2})l^{-2}\right] \leq K'(x)l^{-1},$$

which completes the proof. □

Recall that

$$\text{SVAG:} \quad \Delta(x) := x_1^{x,0} - x, \qquad \text{SDE:} \quad \widetilde{\Delta}(x) := \widetilde{X}_1^{x,0} - x. \tag{26}$$

**Lemma C.2.** *Suppose $u^1, \ldots, u^k \in G^4$ uniformly, that is, $u^1, \ldots, u^k \in G$ and there's a single $K_0 \in G$ such that $\left|\frac{\partial^s u}{\partial x_{(i_1)}, \ldots x_{(i_j)}}(x)\right| \leq K_0(x)$, for $s = 1, 2, 3, 4$ and $i_j \in \{1, 2, \ldots, d\}, j \in \{1, \ldots, s\}$. Let assumption (i),(ii) in Thm. C.1 hold and $K_1(x), K_2(x)$ be such functions. Then, there exists some $K \in G$, independent of $l, r$, such that*

$$\left|\mathbb{E}u^r(x_1^{x,0}) - \mathbb{E}u^r(\widetilde{X}_1^{x,0})\right| \leq K(x)l^{-2}$$

*Proof.* W.L.O.G, we can assume $K_0(x) = \kappa_{0,1}(1 + |x|^{2\kappa_{0,2}}) \leq K_0(x)^2$, for $\kappa_{0,1} > 0, \kappa_{0,2} \in \mathbb{N}^+$, thus for $\alpha \in [0, 1]$ and $x, y \in \mathbb{R}^d$, we have $K_0((1 - \alpha)x + \alpha y) \leq \max(K_0(x), K_0(y)) \leq K_0(x) + K_0(y)$. We also assume $\mathbb{E}K_0(x_1^{x,0})^2 \leq K_2^2(x)$.

Using Taylor's theorem with the Lagrange form of the remainder, we have for any $j \in \{1, \ldots, k\}$,

$$u^r(x_1^{x,0}) - u^r(\widetilde{X}_1^{x,0}) = \sum_{s=1}^{3} \frac{1}{s!} \sum_{i_1, \ldots, i_j=1}^{d} \prod_{j=1}^{s} [\Delta_{(i_j)}(x) - \widetilde{\Delta}_{(i_j)}(x)] \frac{\partial^s u^r}{\partial x_{(i_1)}, \ldots x_{(i_j)}}(x)$$

$$+ \frac{1}{4!} \sum_{i_1, \ldots, i_4=1}^{d} \left[ \frac{\partial^4 u^r}{\partial x_{(i_1)}, \ldots x_{(i_4)}}(x + a\Delta(x)) \prod_{j=1}^{4} \Delta_{(i_4)}(x) \right]$$

$$- \frac{1}{4!} \sum_{i_1, \ldots, i_4=1}^{d} \left[ \frac{\partial^4 u^r}{\partial x_{(i_1)}, \ldots x_{(i_4)}}(x + a\Delta(x)) \prod_{j=1}^{4} \Delta_{(i_4)}(x) \right]$$

where $a, \widetilde{a} \in [0, 1]$.

Taking expectations over the first term, using assumption (i) of Thm. C.1, we get

$$\left| \mathbb{E}\left[ \sum_{s=1}^{3} \frac{1}{s!} \sum_{i_1, \ldots, i_j=1}^{d} \prod_{j=1}^{s} [\Delta_{(i_j)}(x) - \widetilde{\Delta}_{(i_j)}(x)] \frac{\partial^s u^r}{\partial x_{(i_1)}, \ldots x_{(i_j)}}(x) \right] \right| \leq l^{-2}\left(\frac{d}{1!} + \frac{d^2}{2!} + \frac{d^3}{3!}\right)K_1(x)K_0(x)$$

Taking expectations over the second term, using assumption (i) of Appendix C.1 and Lemma D.1, we get

$$\left| \mathbb{E}\left[ \frac{1}{4!} \sum_{i_1,\ldots,i_j=1}^{d} \left[ \frac{\partial^4 u^r}{\partial x_{(i_1)},\ldots x_{(i_4)}}(x+a\Delta(x)) \prod_{j=1}^{4} \Delta_{(i_4)}(x) \right] \right] \right|$$

$$\leq \frac{1}{4!} \sum_{i_1,\ldots,i_4=1}^{d} \mathbb{E}\left| \frac{\partial^4 u^r}{\partial x_{(i_1)},\ldots x_{(i_4)}}(x+a\Delta(x)) \right| \left| \prod_{j=1}^{4} \Delta_{(i_4)}(x) \right|$$

$$\leq \frac{1}{4!} \sum_{i_1,\ldots,i_4=1}^{d} \sqrt{ \mathbb{E}\left| \frac{\partial^{(\alpha+1)} u^r}{\partial x_{(i_1)},\ldots x_{(i_4)}}(x+a\Delta(x)) \right|^2 \mathbb{E}\left| \prod_{j=1}^{4} \Delta_{(i_4)}(x) \right|^2 }$$

$$\leq \frac{1}{4! l^2} \sum_{i_1,\ldots,i_4=1}^{d} \sqrt{ \mathbb{E}\left| K_0(x) + K_0(x_1^{x,0}) \right|^2 K_1(x)^2 }$$

Note that by assumption (ii) of Appendix C.1, we have

$$\mathbb{E}\left| K_0(x) + K_0(x_1^{x,0}) \right|^2 \leq 2K_0(x)^2 + 2\mathbb{E}K_0(x_1^{x,0})^2 \leq 2K_0(x)^2 + 2K_2(x)^2 \leq (2K_0(x) + 2K_2(x))^2.$$

Thus,

$$\left| \mathbb{E}\left[ \frac{1}{4!} \sum_{i_1,\ldots,i_4=1}^{d} \left[ \frac{\partial^4 u^r}{\partial x_{(i_1)},\ldots x_{(i_4)}}(x+a\Delta(x)) \prod_{j=1}^{4} \Delta_{(i_4)}(x) \right] \right] \right|$$

$$\leq l^{-2} \frac{d^4}{4!}(2K_0(x) + 2K_2(x))K_1(x)$$

We can deal with the third term similarly to the second term and thus we conclude

$$|\mathbb{E}u(x_1^{x,0}) - \mathbb{E}u(\widetilde{X}_1^{x,0})| \leq K(x)l^{-2}$$

$\square$

## C.2 One-step approximation

**Lemma 4.6.** *Define the one-step increment of the Itô SDE as $\widetilde{\Delta}(x) = \widetilde{X}_1^{x,0} - x$. Then we have*

*(i)* $\mathbb{E}\widetilde{\Delta}(x) = -\frac{\eta}{l}\nabla\mathcal{L}(x) + \mathcal{O}(l^{-2})$,      *(ii)* $\mathbb{E}\widetilde{\Delta}(x)\widetilde{\Delta}(x)^\top = \frac{\eta^2}{l}\Sigma(x) + \mathcal{O}(l^{-2})$,

*(ii)* $\mathbb{E}\widetilde{\Delta}(x)^{\otimes 3} = \mathcal{O}(l^{-2})$,      *(iv)* $\sqrt{\mathbb{E}|\widetilde{\Delta}(x)^{\otimes 4}|^2} = \mathcal{O}(l^{-2})$.

*Proof.* To obtain (i)-(iii), we simply apply Lem. D.2 with $\psi(z) = \prod_{j=1}^{s}(z_{(i_j)} - x_{(i_j)})$ for $s = 1, 2, 3$ and $i_j \in \{1,\ldots,d\}$ respectively. (iv) is due to Lemma D.1. $\square$

Next, we estimate the moments of the SVAG iterations below.

**Lemma 4.7.** *Define the one-step increment of SVAG as $\Delta(x) = x_1^{x,0} - x$. Then we have*

*(i)* $\mathbb{E}\Delta(x) = -\frac{\eta}{l}\nabla\mathcal{L}(x)$,

*(ii)* $\mathbb{E}\Delta(x)\Delta(x)^\top = \frac{\eta^2}{l}\Sigma(x) + \frac{\eta^2}{l^2}\nabla\mathcal{L}(x)\nabla\mathcal{L}(x)^\top = \frac{\eta^2}{l}\Sigma(x) + \mathcal{O}(l^{-2})$,

*(iii)* $\mathbb{E}\Delta(x)^{\otimes 3} = \frac{\eta^3}{l^2}\frac{3-l^{-1}}{2}\Lambda(x) + \frac{\eta^3}{l^3}\left(3\overline{\nabla\mathcal{L}(x) \otimes \Sigma(x)} + \nabla\mathcal{L}(x)^{\otimes 3}\right) = \mathcal{O}(l^{-2})$

*(iv)* $\sqrt{\mathbb{E}|\Delta(x)^{\otimes 4}|^2} = \mathcal{O}(l^{-2})$,

*where $\Lambda(x) := \mathbb{E}(\nabla\mathcal{L}_{\gamma_1}(x) - \nabla\mathcal{L}(x))^{\otimes 3}$, and $\overline{\mathcal{T}}$ denotes the symmetrization of tensor $\mathcal{T}$, i.e., $\overline{\mathcal{T}}_{ijk} = \frac{1}{6}\sum_{i',j',k'} \mathcal{T}_{i'j'k'}$, where $i', j', k'$ sums over all permutation of $i, j, k$.*

*Proof.* Recall $\Delta(x) = -\frac{\eta}{l}\nabla\mathcal{L}_{\bar{\gamma}}(x)$, where $\mathcal{L}_{\bar{\gamma}}(x) = \frac{1+\sqrt{2l-1}}{2}\mathcal{L}_{\gamma_1}(x) + \frac{1-\sqrt{2l-1}}{2}\mathcal{L}_{\gamma_2}(x)$. Taking expectations, (i) and (ii) are immediate. Note $|\Delta(x)| = O(l^{-0.5})$, (iv) also holds.

Below we show (iii). For convenience, we denote $\sqrt{2l-1}$ by $c$, $\nabla = \nabla\mathcal{L}(x)$, $\widetilde{\nabla}_i = \nabla\mathcal{L}_{\gamma_i}(x) - \nabla\mathcal{L}(x)$, for $i = 1, 2$ and $\widetilde{\nabla} = \frac{1+c}{2}\widetilde{\nabla}_1 + \frac{1-c}{2}\widetilde{\nabla}_2 = \nabla\mathcal{L}_\gamma(x) - \nabla\mathcal{L}(x)$. We have

$$\mathbb{E}\frac{l^3}{\eta^3}\Delta(x)^{\otimes 3} = \mathbb{E}(\widetilde{\nabla} + \nabla)^{\otimes 3}$$
$$= \mathbb{E}\widetilde{\nabla}^{\otimes 3} + 3\mathbb{E}\overline{\widetilde{\nabla} \otimes \widetilde{\nabla} \otimes \nabla} + \nabla^{\otimes 3} \quad (\mathbb{E}\widetilde{\nabla} = 0)$$
$$= \mathbb{E}\widetilde{\nabla}^{\otimes 3} + 3\mathbb{E}\overline{\Sigma \otimes \nabla} + \nabla^{\otimes 3}$$
$$= \frac{3l-1}{2}\Lambda(x) + 3\mathbb{E}\overline{\Sigma \otimes \nabla} + \nabla^{\otimes 3},$$

where the last step is because

$$\mathbb{E}\widetilde{\nabla}^{\otimes 3}(x) = \mathbb{E}(\frac{1+c}{2}\widetilde{\nabla}^1 + \frac{1-c}{2}\widetilde{\nabla}^2)^{\otimes 3}(x)$$
$$= [((\frac{1+c}{2})^3 + (\frac{1-c}{2})^3)]\mathbb{E}(\nabla\mathcal{L}_{\gamma_1}(x) - \nabla\mathcal{L}(x))^{\otimes 3}$$
$$= [((\frac{1+c}{2})^3 + (\frac{1-c}{2})^3)]\Lambda(x)$$
$$= \frac{3l-1}{2}\Lambda(x).$$

$\square$

# D   Auxiliary results for the proof of Thm. 4.3

**Lemma D.1.** *Let $\alpha \geq 1$, there exists a $K \in G$, independent of $l$, such that*

$$\mathbb{E}\prod_{j=1}^{\alpha}\left|\widetilde{\Delta}_{(i_j)}\right| \leq K(x)l^{-\frac{\alpha}{2}}.$$

*where $i_j \in \{1, \dots, d\}$ and $C > 0$ is independent of $l$.*

*Proof.* We have

$$\mathbb{E}|\widetilde{\Delta}(x)|^\alpha \leq 2^{\alpha-1}\mathbb{E}\left|\int_0^{\frac{\eta}{l}}\nabla\mathcal{L}(X_s^{x,0})ds\right|^\alpha + 2^{\alpha-1}\mathbb{E}\left|\int_0^{\frac{\eta}{l}}\Sigma^{0.5}(X_s^{x,0})dW_s\right|^\alpha$$
$$\leq 2^{\alpha-1}(\frac{\eta}{l})^{\alpha-1}\int_0^{\frac{\eta}{l}}\mathbb{E}|\nabla\mathcal{L}(X_s^{x,0})|^\alpha ds + 2^{\alpha-1}\left|\int_0^{\frac{\eta}{l}}\Sigma^{0.5}(X_s^{x,0})dW_s\right|^\alpha$$

Using Cauchy-Schwarz inequality, Itô's isometry, we get

$$\mathbb{E}\left|\int_0^{\frac{\eta}{l}}\sigma(X_s^{x,0})dW_s\right|^\alpha \leq \left(\mathbb{E}\left|\int_0^{\frac{\eta}{l}}\sigma(X_s^{x,0})dW_s\right|^{2\alpha}\right)^{1/2}$$
$$\leq C(\frac{\eta}{l})^{\alpha-1/2}\left(\int_0^{\frac{\eta}{l}}\mathbb{E}|\sigma(X_s^{x,0})|^{2\alpha}ds\right)^{1/2}$$
$$= O(l^{-\alpha/2})$$

where $C$ depends only on $\alpha$. Now, using the linear growth condition (4.3 (ii)) and the moment estimates in Theorem 19 in (Li et al., 2019a), we obtain the result. $\square$

We prove the following Itô-Taylor expansion, which is slightly different from Lemma 28 in (Li et al., 2019a).

**Lemma D.2.** *Let $\psi : \mathbb{R}^d \to \mathbb{R}$ be a sufficiently smooth function.*

*Suppose that $b, \sigma \in G^3$, and $X_t^{x,0}$ is the solution of the following SDE, with $X_0^{x,0} = x$.*

$$dX_t = b(X_t)dt + \sigma(X_t)dW_t.$$

*Then we have*

$$\mathbb{E}\psi(X_\eta^{x,0}) = \psi(x) + \eta b(x)^\top \nabla \psi(x) + \eta \mathrm{Tr}\left[\nabla^2 \psi \cdot \sigma^2\right](x) + \mathcal{O}(\eta^2).$$

*That is, there exists some function $K \in G$ such that*

$$|\mathbb{E}\psi(X_\eta^{x,0}) - \psi(x) - \eta b(x)^\top \nabla \psi(x) - \eta \mathrm{Tr}\left[\nabla^2 \psi \cdot \sigma^2\right](x)| \le K(x)\eta^2.$$

*Proof.* We define operator $A_{1,\epsilon}\psi := b^\top \nabla \psi$, $A_{2,\epsilon}\psi := \frac{1}{2}\mathrm{Tr}\left[\nabla^2 \psi \cdot \sigma^2\right]$.

Using Itô's formula, we have

$$\mathbb{E}\psi(X_\eta^{x,0}) = \psi(x) + \int_0^\eta \mathbb{E}A_{1,\epsilon}\psi(X_s^{x,0})ds + \int_0^\eta \mathbb{E}A_{2,\epsilon}\psi(X_s^{x,0})ds$$

By further application of the above formula to $\mathbb{E}A_{1,\epsilon}\psi$ and $\mathbb{E}A_{2,\epsilon}\psi$, we have

$$\mathbb{E}\psi(X_\eta^{x,0}) = \psi(x) + \eta A_{1,\epsilon}\psi(x) + \eta A_{2,\epsilon}\psi(x)$$
$$+ \int_0^\eta \int_0^s \mathbb{E}((A_{1,\epsilon} + A_{2,\epsilon})(A_{1,\epsilon} + A_{2,\epsilon}))\psi(X_v^{x,0})dvds$$

Taking expectations of the above, it remains to show that each of the terms is $\mathcal{O}(\eta^2)$. This follows immediately from the assumption that $b, \sigma \in G^3$ and $\psi \in G^4$. Indeed, observe that all the integrands have at most 3 derivatives in $b_0, b_1, \sigma_0$ and 4 derivatives in $\psi$, which by our assumptions all belong to $G$. Thus, the expectation of each integrand is bounded by $\kappa_1(1 + \sup_{t \in [0,\eta]} \mathbb{E}|X_t^{x,0}|^{2\kappa_2})$ for some $\kappa_1, \kappa_2$, which by Theorem 19 in (Li et al., 2019a) must be finite. Thus, the expectations of the other integrals are $\mathcal{O}(\eta^2)$ by the polynomial growth assumption and moment estimates in Theorem 19 in (Li et al., 2019a). □

We also prove a general moment estimate for the SVAG iterations Equation (5).

**Lemma D.3.** *Let $\{x_k : k \ge 0\}$ be the generalized SVAG iterations defined in Equation (5). Suppose*

$$|\nabla \mathcal{L}_\gamma(x)| \le L_\gamma(1 + |x|), \quad \forall x \in \mathbb{R}^d, \gamma$$

*for some random variable $L_\gamma > 0$ with all moments bounded, i.e., $\mathbb{E}L_\gamma^k < \infty$, for $k \in \mathbb{N}$. Then, for fixed $T > 0$ and any $m \ge 1$, $\mathbb{E}|x_k|^{2m}$ exists and is uniformly bounded in $l$ and $k = 0, \ldots, N \equiv \lfloor lT/\eta \rfloor$.*

*Proof.* Recall that $\nabla \mathcal{L}_{\bar{\gamma}_k}^l(x) = \frac{1+\sqrt{2l-1}}{2}\nabla \mathcal{L}_{\gamma_{k,1}}(x) + \frac{1-\sqrt{2l-1}}{2}\nabla \mathcal{L}_{\gamma_{k,2}}(x)$, thus there exists random variable $L_{\bar{\gamma}}'$ with all moments bounded and $|\nabla \mathcal{L}_{\bar{\gamma}_k}^l(x)|^2 \le l(L_{\bar{\gamma}}')^2(1 + |x|^2)$. We further define $L := \mathbb{E}L_\gamma$, and thus $|\langle \mathbb{E}\nabla \mathcal{L}_{\bar{\gamma}_k}^l(x_k), x_k \rangle| \le 2L(1 + |x|^2)$.

For each $k \ge 0$, we have

$$|x_{k+1}|^{2m} \le (1 + |x_{k+1}^2|)^m = \left|1 + |x_k|^2 - 2\frac{\eta}{l}\left\langle \nabla \mathcal{L}_{\bar{\gamma}_k}^l(x_k), x_k \right\rangle + \frac{\eta^2}{l^2}|\nabla \mathcal{L}_{\bar{\gamma}_k}^l(x_k)|\right|^m$$

$$= (1 + |x_k|)^{2m} - 2m\frac{\eta}{l}\left\langle \nabla \mathcal{L}_{\bar{\gamma}_k}^l(x_k), x_k \right\rangle(1 + |x_k|^2)^{m-1} + \frac{1}{l} \cdot O((|x_k|^2 + 1)^m)$$

Hence, if we let $a_k := \mathbb{E}(1 + |x_k|^2)^m$, we have

$$a_{k+1} = a_{k+1} - 2m\frac{\eta}{l}\left\langle \mathbb{E}\nabla \mathcal{L}_{\bar{\gamma}_k}^l(x_k), x_k \right\rangle(1 + |x_k|^2)^{m-1} + \frac{1}{l} \cdot O((|x_k|^2 + 1)^m) \le (1 + \frac{C}{l})a_k$$

where $C > 0$ are independent of $l$ and $k$, which immediately implies, for all $k = 0, \ldots, \lfloor \frac{lT}{\eta} \rfloor$,

$$a_k \le (1 + C/l)^k a_0 \le (1 + C/l)^{lT/\eta}a_0 \le e^{C\frac{CT}{\eta}}a_0.$$

□

# E Omitted proofs in Section 5

In this section, we provide the missing proofs in Section 5, including Theorem 5.5, Theorem 5.6 and the counterpart of Theorem E.4 between 1st order SDE (2) and 2nd order SDE (32), which is Theorem E.1. We also provide the derivation of properties for scale invariant functions in Appendix E.4.

## E.1 Proof of Theorem 5.6

*Proof of Theorem 5.6.* Suppose $(C, \kappa)$-LSI hold, similar to Equation (11), we have

$$(2 - \kappa\eta\lambda)\lambda R_\infty^{\kappa B, \kappa\eta} = \kappa\eta(G_\infty^{\kappa B, \kappa\eta} + N_\infty^{\kappa B, \kappa\eta}), \tag{27}$$

$$(2 - \eta\lambda)\lambda R_\infty^{B, \eta} = \eta(G_\infty^{B, \eta} + N_\infty^{B, \eta}), \tag{28}$$

Thus combining (27), (28) and (8), we have

$$\kappa(G_\infty^{\kappa B, \kappa\eta} + N_\infty^{\kappa B, \kappa\eta}) = (2 - \kappa\eta\lambda)\frac{\lambda}{\eta}R_\infty^{\kappa B, \kappa\eta} \leq (2 - \eta\lambda)\frac{\lambda}{\eta}CR_\infty^{B, \eta} = C(N_\infty^{B, \eta} + G_\infty^{B, \eta}).$$

Applying (13) again, we have

$$\kappa G_\infty^B + N_\infty^B \leq C\kappa(G_\infty^{\kappa B, \kappa\eta} + N_\infty^{\kappa B, \kappa\eta}) \leq C^2(N_\infty^{B, \eta} + G_\infty^{B, \eta}).$$

Therefore we conclude that $\kappa \leq C^2(1 + \frac{N_\infty^{B, \eta}}{G_\infty^{B, \eta}})$.

$\square$

## E.2 Proof of Theorem 5.5

*Proof.* Suppose $(C, \kappa)$-LSI hold, by (8), we have

$$CG_\infty^{\kappa B, \kappa\eta} + C\kappa N_\infty^{\kappa B, \kappa\eta} \geq G_\infty^{B, \eta} + N_\infty^{B, \eta}.$$

By (27), (28) and (8), we have

$$G_\infty^{B, \eta} + N_\infty^{B, \eta} = 2\lambda\frac{B}{\eta}R_\infty^{B, \eta} \geq \frac{2}{C}\lambda\frac{B}{\eta}R_\infty^{\kappa B, \kappa\eta} = \frac{\kappa}{C}(G_\infty^{\kappa B, \kappa\eta} + N_\infty^{\kappa B, \kappa\eta}). \tag{29}$$

Rearranging things, we have

$$N_\infty^{\kappa B, \kappa\eta} \geq \frac{\kappa - C^2}{\kappa(C^2 - 1)}G_\infty^{\kappa B, \kappa\eta} \geq ((1 - \frac{1}{\kappa})\frac{1}{C^2 - 1} - \frac{1}{\kappa})G_\infty^{\kappa B, \kappa\eta}.$$

$\square$

## E.3 Necessary condition for $C$-closeness between 1st order and 2nd order SDE approximation

In this section we will present a necessary condition for $C$-closeness between 1st order and 2nd order SDE approximation, similar to that betweeen 1st order approximation and SGD. The key observation is that the missing second order term $\eta G_\infty$ in 1st order SDE, also appears in the 2nd order SDE, as it does for SGD. Thus we can basically apply the same analysis and show the similar conclusion (Theorem E.4).

Below we first recap the notion of 1st and 2nd order SDE approximation with weight decay (i.e., $\ell_2$ regularization). We first define $\mathcal{L}'_\gamma(X) = \mathcal{L}_\gamma(X) + \frac{\lambda}{2}|X|^2$ and the SGD dynamics (30) can be written by

$$x_{k+1} = x_k - \eta\nabla\mathcal{L}'_\gamma(x_k) = x_k - \eta\nabla\left(\mathcal{L}_{\gamma_k}(X_t) + \frac{\lambda}{2}|X_t|^2\right) \tag{30}$$

Below we recap the 1st and 2nd order SDE approximation:

- 1st order SDE approximation (with $\overline{\Sigma} = \eta\Sigma$):

$$\mathrm{d}X_t = -\nabla\big(\mathcal{L}(X_t) + \frac{\lambda}{2}|X_t|^2\big)\mathrm{d}t + (\eta\Sigma)^{1/2}(X_t)\mathrm{d}W_t \tag{31}$$

- 2nd order SDE approximation:

$$\mathrm{d}X_t = -\nabla\big(\mathcal{L}'(X_t) + \frac{\eta}{4}|\nabla\mathcal{L}'(X_t)|^2\big)\mathrm{d}t + (\eta\Sigma)^{1/2}(X_t)\mathrm{d}W_t \tag{32}$$

**Theorem E.1** (Theorem 9 in Li et al. (2019a))**.** *(32) is an order-2 weak approximation of SGD (1).*

We first prove a useful lemma.

**Lemma E.2.** *Suppose $\mathcal{L}$ is scale invariant, then for any $X \in \mathbb{R}^d$, $X \neq 0$,*

$$X^\top\nabla^2\mathcal{L}(X)\nabla\mathcal{L}(X) = \frac{1}{2}X^\top\nabla(\|\nabla\mathcal{L}\|_2^2) = -\|\nabla\mathcal{L}(X)\|_2^2.$$

*Proof.* By chain rule, we have

$$
\begin{aligned}
&X^\top\nabla(\|\nabla\mathcal{L}\|_2^2)\\
&= \lim_{t\to 0}\frac{\|\nabla\mathcal{L}((1+t)X)\|_2^2 - \|\nabla\mathcal{L}(X)\|_2^2}{t}\\
&= \lim_{t\to 0}\frac{(1+t)^{-2} - 1}{t}\|\nabla\mathcal{L}(X)\|_2^2 \quad \text{(by scale invariance)}\\
&= -2\|\nabla\mathcal{L}(X)\|_2^2
\end{aligned}
$$

$\square$

**Definition E.3** (*C*-closeness)**.** We use $\overline{R}'_\infty := \lim_{t\to\infty} E|X_t|^2, \overline{G}'_\infty := \lim_{t\to\infty}\mathbb{E}|\nabla\mathcal{L}(X_t)|^2, \overline{N}'_\infty := \lim_{t\to\infty}\mathbb{E}[\mathrm{Tr}[\overline{\Sigma}(X_t)] = \lim_{t\to\infty}\mathbb{E}[\mathrm{Tr}[\eta\Sigma(X_t)]$ to denote the limiting squared norm, gradient norm and trace of covariance for SDE (32). (We assume both $X_t$ converge to their equilibrium so the limits exist). We say the two equilibriums of 1st order SDE approximation (31) and 2nd order SDE approximation (34) are *C-close* to each other iff

$$\frac{1}{C} \le \frac{\overline{R}_\infty}{\overline{R}'_\infty}, \frac{\overline{G}_\infty}{\overline{G}'_\infty}, \frac{\overline{N}_\infty}{\overline{N}'_\infty} \le C. \tag{33}$$

The following theorem is an analog of Theorem 5.2.

**Theorem E.4.** *If the equilibriums of (31) and (32) exist and are C-close for some $C > 0$, then*

$$\eta \le \frac{\overline{N}_\infty}{\overline{G}_\infty}\big(C^2(1 + \frac{\eta\lambda}{2}) - 1\big) \approx \frac{\overline{N}_\infty}{\overline{G}_\infty}\big(C^2 - 1\big),$$

*where $\lambda$ is usually of scale $10^{-4}$ in practice and thus can be omitted when calculating upper bound.*

*Proof.* Since $\mathcal{L}$ is scale-invariant, so $\nabla\mathcal{L}(X)^\top X = 0$, which implies $|\nabla\mathcal{L}'(X)|^2 = |\nabla\mathcal{L}(X)|^2 + \lambda^2|X|^2$. Plug in $\mathcal{L}'$, we have

$$\mathrm{d}X_t = -\Big(\nabla\mathcal{L}(X_t) + \frac{\eta}{2}\nabla^2\mathcal{L}(X_t)\nabla\mathcal{L}(X_t)\Big)\mathrm{d}t + (\eta\Sigma)^{1/2}(X_t)\mathrm{d}W_t - \lambda(1 + \frac{\eta\lambda}{2})X_t\mathrm{d}t. \tag{34}$$

Applying Itô' lemma, we have

$$
\begin{aligned}
\mathrm{d}|X_t|^2 =& 2\langle X_t, \mathrm{d}X_t\rangle + \langle \mathrm{d}X_t, \mathrm{d}X_t\rangle\\
=& -2\lambda(1 + \frac{\eta\lambda}{2})|X_t|^2\mathrm{d}t - \eta\langle X_t, \nabla^2\mathcal{L}(X_t)\nabla\mathcal{L}(X_t)\rangle\mathrm{d}t + \cancel{2\langle X_t, \Sigma^{1/2}(X_t)\mathrm{d}W_t\rangle} - \cancel{2\langle X_t, \nabla\mathcal{L}(X_t)\rangle\mathrm{d}t}\\
& + \mathrm{Tr}[\Sigma(X_t)]\mathrm{d}t\\
=& -2\lambda(1 + \frac{\eta\lambda}{2})|X_t|^2\mathrm{d}t - \eta\langle X_t, \nabla^2\mathcal{L}(X_t)\nabla\mathcal{L}(X_t)\rangle\mathrm{d}t + \mathrm{Tr}[\eta\Sigma(X_t)]\mathrm{d}t \quad \text{(by Corollary E.8)}\\
=& -2\lambda(1 + \frac{\eta\lambda}{2})|X_t|^2\mathrm{d}t + \big(\eta\,\mathrm{Tr}[\Sigma(X_t)] + \eta|\nabla\mathcal{L}(X_t)|^2\big)\mathrm{d}t \quad \text{(by Lemma E.2)}
\end{aligned}
$$

$$\tag{35}$$

Thus

$$\frac{d\mathbb{E}[|X_t|^2]}{dt} = -2\lambda(1 + \frac{\eta\lambda}{2})\mathbb{E}[|X_t|^2] + \mathbb{E}[\eta \operatorname{Tr}[\Sigma(X_t)] + \eta|\nabla\mathcal{L}(X_t)|^2]. \tag{36}$$

Suppose $X_t$ is samples from the equilibrium, we have $\frac{d\mathbb{E}[|X_t|^2]}{dt} = 0$ and

$$(2 + \eta\lambda)\lambda\overline{R}'_\infty = \eta\overline{G}'_\infty + \overline{N}'_\infty. \tag{37}$$

If we compare Equation (37) to Equations (11) and (12) (we recap them below), it's quite clear 2nd order is much closer to SGD in terms of the relationship between $R_\infty$, $G_\infty$ and $N_\infty$. Thus 1st and 2nd order SDE approximation won't be $C$-close if $\frac{\overline{N}_\infty}{\overline{G}_\infty}$ is larger than some constant for the exact same reason that 1st SDE is not $C$-close to SGD.

$$(2 - \eta\lambda)\lambda R_\infty = \eta G_\infty + \eta N_\infty, \tag{11}$$
$$2\lambda\overline{R}_\infty = \quad + \overline{N}_\infty. \tag{12}$$

In detail, by combining (37), (12) and (33), we have

$$\eta\overline{G}'_\infty + \overline{N}'_\infty = (2 + \eta\lambda)\lambda\overline{R}'_\infty \le (2 + \eta\lambda)\lambda C\overline{R}_\infty = (1 + \frac{\eta\lambda}{2})C\overline{N}_\infty.$$

Applying (33) again, we have

$$\eta\overline{G}_\infty + \overline{N}_\infty \le C(\eta\overline{G}'_\infty + \overline{N}'_\infty) \le C^2(1 + \frac{\eta\lambda}{2})\overline{N}_\infty \le C^3(1 + \frac{\eta\lambda}{2})\overline{N}'_\infty,$$

which imples $\eta \le \left(C^2(1 + \frac{\eta\lambda}{2}) - 1\right)\min\{\frac{\overline{N}_\infty}{\overline{G}_\infty}, \frac{\overline{N}'_\infty}{\overline{G}'_\infty}\}$. $\qquad\square$

**Remark E.5.** *Kunin et al. (2020) derived a similar equation to Equation (35) in Appendix F of their paper.*

### E.4 Properties of Scale Invariance Function

These properties are proved in Arora et al. (2019b). We include them here for self-containedness.

**Definition E.6** (Scale Invariance). We say $\mathcal{L} : \mathbb{R}^d \to R$ is *scale invariant* iff $\forall x \in \mathbb{R}^d, x \ne 0$ and $\forall c > 0$, it holds that
$$\mathcal{L}(cx) = \mathcal{L}(x).$$

We have the following two properties.

**Lemma E.7.** *If $\mathcal{L}$ is scale invariant, then $\forall x \in \mathbb{R}^d/\{0\}$, we have*

(1) $\langle x, \nabla\mathcal{L}(x)\rangle = 0$.
(2) $\forall c > 0$, $c\nabla\mathcal{L}(cx) = \nabla\mathcal{L}(x)$.
(3) $\forall c > 0$, $c^2\nabla^2\mathcal{L}(cx) = \nabla^2\mathcal{L}(x)$.

*Proof.* For (1), by chain rule, we have $\langle x, \nabla\mathcal{L}(x)\rangle = \lim_{t\to 0}\frac{\mathcal{L}((1+t)x) - \mathcal{L}(x)}{t} = 0$.

For (2), for any $v \in \mathbb{R}^d$, again by chain rule, we have

$$\langle v, c\nabla\mathcal{L}(cx)\rangle = \langle cv, \nabla\mathcal{L}(cx)\rangle = \lim_{t\to 0}\frac{\mathcal{L}(cx + cvt) - \mathcal{L}(cx)}{t} = \lim_{t\to 0}\frac{\mathcal{L}(x + vt) - \mathcal{L}(x)}{t} = \langle v, \nabla\mathcal{L}(x)\rangle.$$

For (3), take gradient on both sides of (2) over $x$ and the proof is completed. $\qquad\square$

Suppose $\mathcal{L}_\gamma$ is a random loss and is scale invariant for every $\gamma$, and we use $\nabla\mathcal{L}(x)$ and $\Sigma(x)$ to denote the expectation and covariance of gradient, then for any $x \in \mathbb{R}^d/\{0\}$, we have the following corollary:

**Corollary E.8.** $\langle x, \nabla\mathcal{L}(x)\rangle = 0$, $x^\top\Sigma(x)x = 0$.

# F Experiments

We use the models from Github Repository: https://github.com/bearpaw/pytorch-classification. For VGG and PreResNet, unless noted otherwise, we modified the model following Appendix C of (Li and Arora, 2020b) so that the network is scale invariant, e.g., fixing the last layer. Such modification doesn't lead to change in performance, as shown in Hoffer et al. (2018). We use Weights & Biases to manage our experiments (Biewald, 2020).

## F.1 Further Verification of SVAG

We verify that SVAG converges for different architectures (including ones without normalization), learning rate schedules, and datasets. We further conclude that for most the standard settings we consider (excluding the use of large batch size in Figures 4 and 14 and GroupNorm on CIFAR-100 in Figure 13), SVAG with large $l$ achieves similar performance to SGD, i.e. SVAG with $l = 1$.

Theorem 4.3 only holds if each step of SGD is a Markov process, which is in part determined by how each example is sampled from the dataset. We describe three common ways that examples can be sampled from the dataset during training.

1. *Random shuffling (RS)*: RS is standard practice in experiments and is the default implementation in PyTorch. A random shuffled order of datapoints is fixed at the start of each epoch, and each sample is drawn in this order. SGD with this sampling scheme can be viewed as a Markov process per epoch, although not per step. We use this method in our experiments, but our theory for SVAG (Section 4) does not cover this sampling method.
2. *Without replacement (WOR)*: WOR requires drawing each sample i.i.d. from the dataset without replacing previously drawn ones. SGD with this sampling scheme can be viewed as a Markov process per step, so our theory for SVAG (Section 4) does cover this case.
3. *With replacement (WR)*: WR requires drawing each sample i.i.d. from the dataset with replacement. SGD with this sampling scheme can be viewed as a Markov process per step, so our theory for SVAG (Section 4) does cover this case.

In Figure 7, we observe that SVAG (including SGD) behaves similarly when using all three of these sampling methods. Therefore, although our theory does not directly apply to the commonly used RS scheme, we can heuristically apply Theorem 4.3 to understand its behavior.

We furthermore note that our findings do not match the conclusion in Smith et al. (2021) that SGD with RS has a different implicit bias compared to WOR and WR. This is possibly because the analysis of Smith et al. (2021) doesn't apply to SGD with commonly used tricks like data augmentation and Batch Normalization. These tricks break the most important property of random shuffling SGD used in the analysis of Smith et al. (2021), that the average of batch loss functions in an epoch is equal to the expected loss function. The other possible reasons for this discrepancy includes:

(1) Their result concerns behavior after a single epoch and our experiments run for hundreds of epochs; (2) Their result holds when $\eta$ is smaller than an unmeasurable constant, so it may be the case that their results do not apply to the constant LR regime we use SVAG in.

### F.1.1 Further Verification of SVAG on more architectures on CIFAR-10

For all CIFAR-10 experiments in this subsection, there are 320 epochs with initial LR $\eta = 0.8$ and 2 LR decays by a factor of 0.1 at epochs 160 and 240 and we use weight decay with $\lambda = 5e{-}4$ and batch size $B = 128$. We also use the standard data augmentation for CIFAR-10: taking a random $32 \times 32$ crop after padding with 4 pixels and randomly performing a horizontal flip.

In Figure 8, we demonstrate that SVAG converges and closely follows SGD for PreResNet32 with BatchNorm (left), PreResNet32 (4x) with BatchNorm (middle) and PreResNet32 with GroupNorm (right).

In Figure 9, we demonstrate that SVAG converges and closely follows SGD for VGG16 without Normalization (left), VGG16 with BatchNorm (middle) and VGG16 with GroupNorm (right).

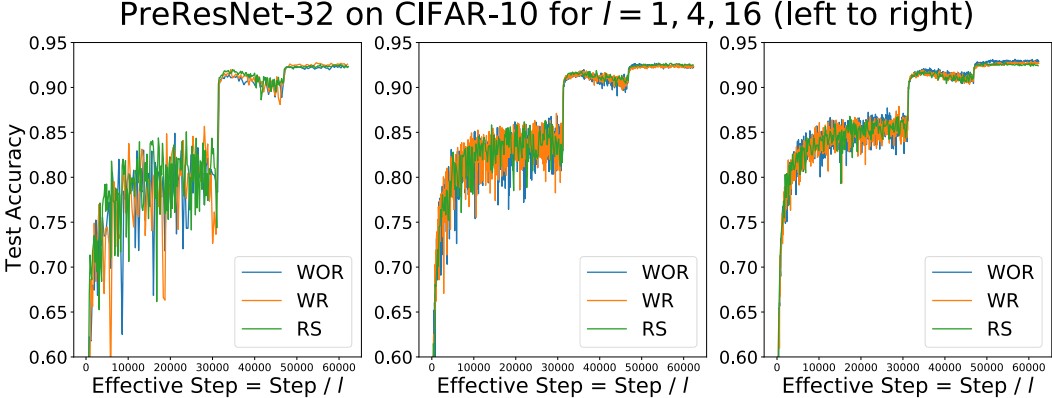

Figure 7: Different sampling methods have little impact on the performance of SVAG (SGD). We compare random shuffling (RS), which is the default implementation in PyTorch, to with replacement (WR) and without replacement (WOR), which Theorem 4.3 applies to. We train for 320 epochs with initial LR $\eta = 0.8$ with 2 LR decays by a factor of 0.1 at epochs 160 and 240, and we use weight decay with $\lambda = 5\mathrm{e}{-4}$ and batch size $B = 128$. Since SVAG takes $l$ smaller steps to simulate the continuous dynamics in $\eta$ time, we plot accuracy against "effective steps" defined as $\frac{\#\text{steps}}{l}$.

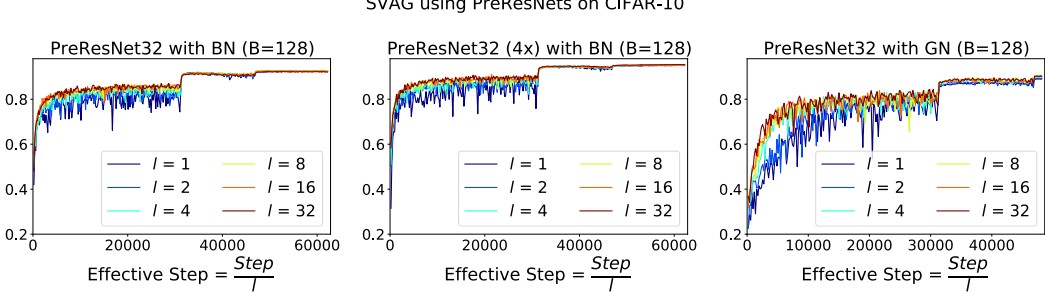

Figure 8: Validation accuracy for PreResNet32 with BatchNorm (left), PreResNet32 (4x) with BatchNorm (middle) and PreResNet32 with GroupNorm (right) during training on CIFAR-10. SVAG converges and closely follows the SGD trajectory in all three cases. Since SVAG takes $l$ smaller steps to simulate the continuous dynamics in $\eta$ time, we plot accuracy against "effective steps" defined as $\frac{\#\text{steps}}{l}$.

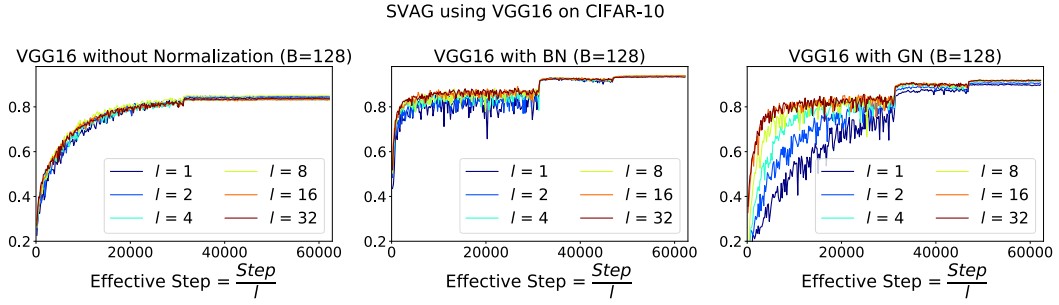

Figure 9: Validation accuracy for VGG16 without Normalization (left), VGG16 with BatchNorm (middle) and VGG16 with GroupNorm (right) during training on CIFAR-10. SVAG converges and closely follows the SGD trajectory in all three cases. Since SVAG takes $l$ smaller steps to simulate the continuous dynamics in $\eta$ time, we plot accuracy against "effective steps" defined as $\frac{\#\text{steps}}{l}$.

### F.1.2 Further Verification of SVAG on more complex LR schedules

We verify that SVAG converges and closely follows the SGD trajectory for networks trained with more complex learning rate schedules. In Figure 11, we use the triangle (i.e., cyclical) learning rate schedule proposed in Smith (2017), visualized in Figure 10. We implement the schedule over 320 epochs of training: we increase the initial learning rate $0.001$ linearly to $0.8$ over $80$ epochs, decay the LR to $0.001$ over the next $80$ epochs, increase the LR to $0.4$ over $80$ epochs, and decay the LR to $0.001$ over the remaining $80$ epochs. As seen in Figure 11, SVAG converges to the SGD trajectory in this setting.

We further test SVAG on the cosine learning rate schedule proposed in Loshchilov and Hutter (2016) with $\eta_{\max} = 0.8$ and $\eta_{\min} = 0.001$ with total training budgets of $160$ epochs. We visualize the schedule in Figure 10. In Figure 12, we see that SVAG converges and closely follows the SGD trajectory, suggesting the SDE (2) can model SGD trajectories with complex learning rate schedules as well.

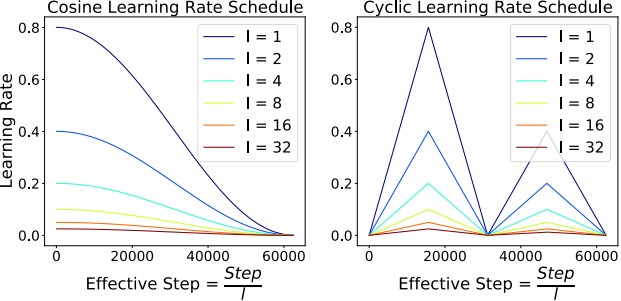

Figure 10: Cosine (left, Loshchilov and Hutter (2016)) and cyclic (right, Smith (2017)) learning rate schedules for different SVAG configurations, plotted against "effective steps" defined as $\frac{\#\text{steps}}{l}$.

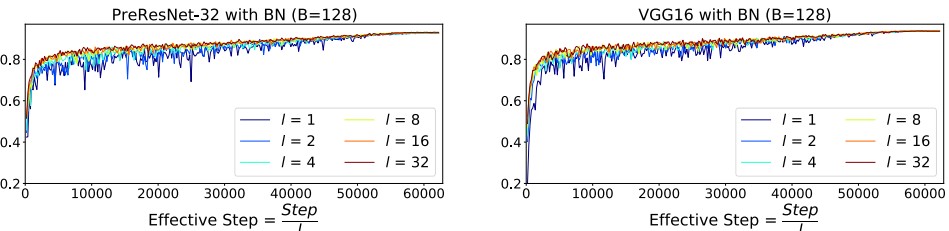

Figure 11: Validation accuracy for PreResNet-32 with $B = 128$ (left) and VGG16 with $B = 128$ (right) during training on CIFAR-10. We use the triangle LR schedule proposed in Smith (2017) with 80 epochs of increase to LR 0.8, 80 epochs of decay to 0, 80 epochs of increase to 0.4 and 80 epochs of decay to 0. The LR schedule is visualized in Figure 10. Since SVAG takes $l$ smaller steps to simulate the continuous dynamics in $\eta$ time, we plot accuracy against "effective steps" defined as $\frac{\#\text{steps}}{l}$.

### F.1.3 Further Verification of SVAG on more datasets (CIFAR-100 and SVHN)

We also verify that SVAG converges on the CIFAR-100 dataset. We set the learning rate to be $0.8$ and decay it by a factor of $0.1$ at epochs $160$ and $240$ with a total budget of $320$ epochs. We use weight decay with $\lambda = 5e{-}4$. We use the standard data augmentation for CIFAR-100: randomly taking a $32 \times 32$ crop from the image after padding with 4 pixels and randomly horizontally flipping the result. We observe that SVAG converges for computationally tractable value of $l$ in Figure 13, but for both GN architectures, the SDE fails to approximate SGD training.

We also verify SVAG on the Street View House Numbers (SVHN) dataset (Netzer et al., 2011). We set the learning rate to $0.8$ for batch size 128 and scale it according to LSR (Definition 2.1) for large batch training. We train for $240$ epochs and decay the learning rate by a factor of $0.1$ once at epoch 200. We use weight decay with $\lambda = 5e{-}4$.

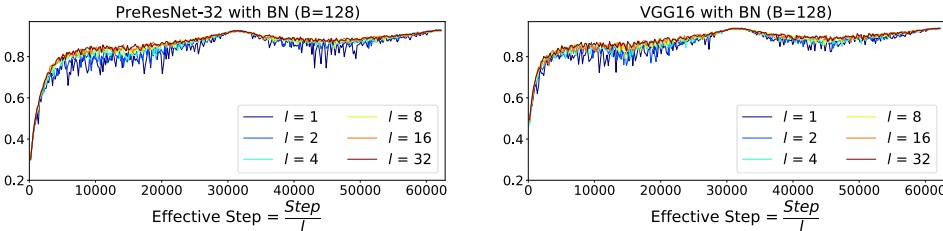

Figure 12: Validation accuracy for PreResNet-32 with $B = 128$ (left) and VGG16 with $B = 128$ (right) during training on CIFAR-10. We use the cosine LR schedule proposed in Loshchilov and Hutter (2016) starting with LR 0.8 and following the cosine curve until the LR becomes infinitesimally small. The LR schedule is visualized in Figure 10. Since SVAG takes $l$ smaller steps to simulate the continuous dynamics in $\eta$ time, we plot accuracy against "effective steps" defined as $\frac{\#\text{steps}}{l}$.

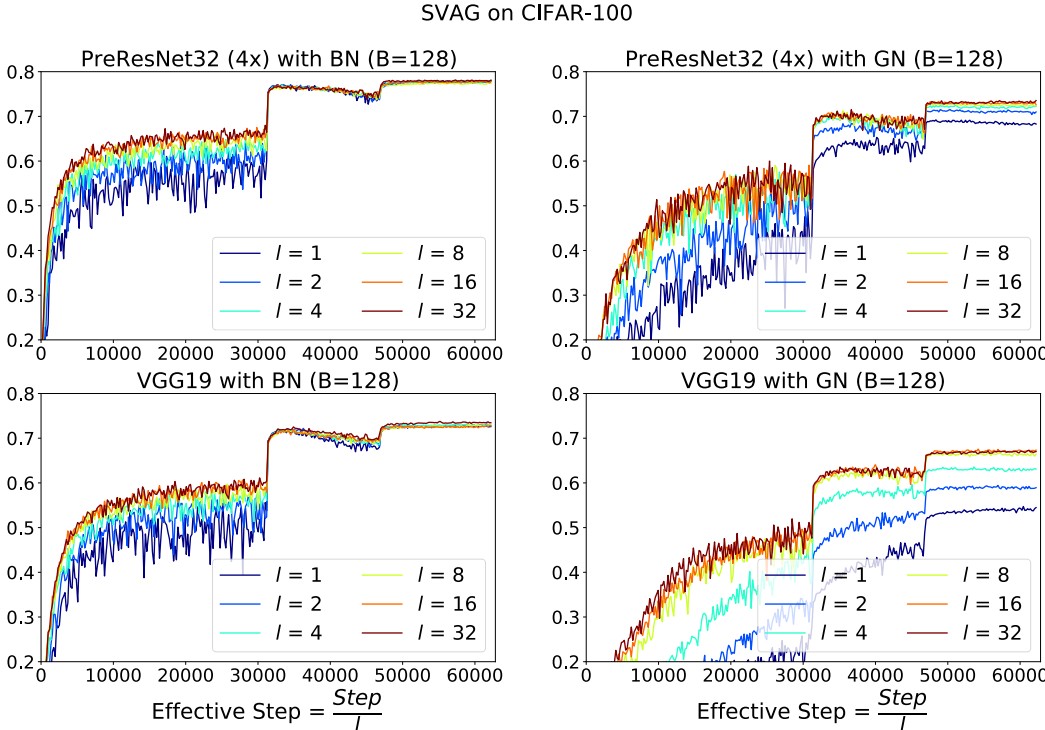

Figure 13: Validation accuracy for wider PreResNet-32 with $B = 128$ using BN (top left) and GN (top right) and for VGG-19 with $B = 128$ using BN (bottom left) and GN (bottom right) trained on CIFAR-100. We train for 320 epochs and decay the LR by 0.1 at epochs 160 and 240. Since SVAG takes $l$ smaller steps to simulate the continuous dynamics in $\eta$ time, we plot accuracy against "effective steps" defined as $\frac{\#\text{steps}}{l}$. Interestingly, we found the performance of BatchNorm out performs GroupNorm and the performance of the latter gets improved when using larger $l$ for SVAG.

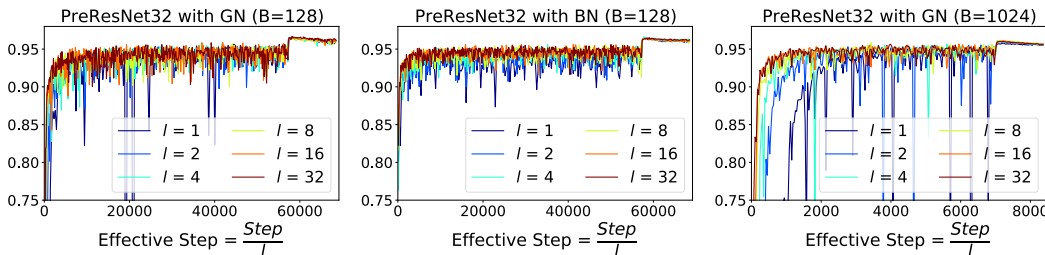

Figure 14: Validation accuracy for PreResNet-32 with $B = 128$ using GN (left) and BN (center) and with $B = 1024$ using GN trained on the SVHN dataset (Netzer et al., 2011). We train for 240 epochs and decay the LR by 0.1 at epoch 200. Since SVAG takes $l$ smaller steps to simulate the continuous dynamics in $\eta$ time, we plot accuracy against "effective steps" defined as $\frac{\#\text{steps}}{l}$.

## F.2 Further Verification of Necessary Condition for LSR

We further verify the necessary condition for LSR (Theorem 5.6) using different architectures and datasets. Figure 15 tests the condition for ResNet-32 and wider PreResNets trained on CIFAR-10. Although our theory requires strict scale-invariance, we find the condition to still be applicable to the standard ResNet architecture (He et al., 2016), ResNet32, likely because most of the network parameters are scale-invariant. Figure 16 tests the condition for wider PreResNets and VGG-19 trained on CIFAR-100. We require the wider PreResNet to achieve reasonable test error, but we note that the larger model made it difficult to straightforwardly train with a larger batch size.

In Figure 15 and Figure 3, $G_t$ and $N_t$ are the empirical estimations of $G_\infty$ and $N_\infty$ taken after reaching equilibrium in the second to last phase (before the final LR decay), where the number of samples (batches) is equal to $\max(200, 50000/B)$, and $B$ is the batch size.

Per the approximated version of Theorem 5.6, i.e., $B^* = \kappa B \lesssim C^2 B N_\infty^B / G_\infty^B$, we use baseline runs with different batch sizes $B$ to report the maximal and minimal predicted critical batch size, defined as the x-coordinate of the intersection of the threshold ($G_t/N_t = C^2$) with the green and blue lines, respectively. Both the green and blue line have slope 1, and thus the x-coordinate of intersection, $B^*$, is the solution of the following equation,

$$\frac{B^*}{B} = \frac{G_t^{B^*}/N_t^{B^*}}{G_t^B/N_t^B}, \text{ where } G_t^{B^*}/N_t^{B^*} = C^2.$$

For all settings, we choose a threshold of $C^2 = 2$, and consider LSR to fail if the final test error exceeds the lowest achieved test error by more than 20% of its value, marked by the red region on the plot. Surprisingly, it turns out the condition in Theorem 5.6 is not only necessary, but also close to sufficient.

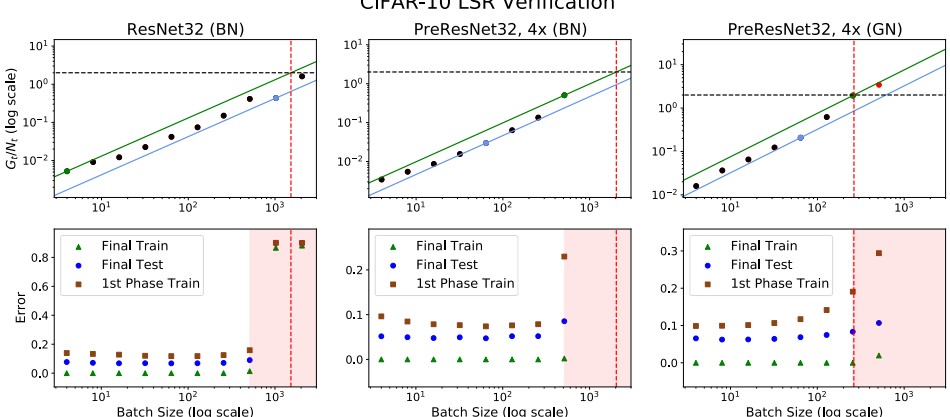

Figure 15: Further verification for our theory on predicting the failure of Linear Scaling Rule. We test if the condition applies to different architectures trained on CIFAR-10. All three settings use the same LR schedule, LR= $0.8$ initially and is decayed by $0.1$ at epoch 250 with 300 epochs total budget. We measure $G_t$ and $N_t$ by averaging their values over the last 50 epochs of the first phase (i.e., from epoch 200 to 250).

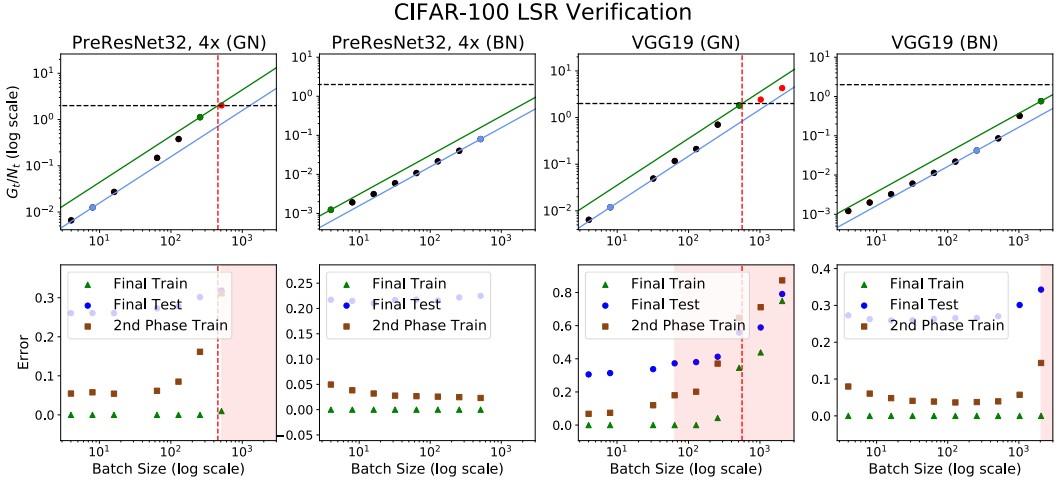

Figure 16: Further verification for our theory on predicting the failure of Linear Scaling Rule. We test if the condition applies to different architectures trained on CIFAR-100. All four settings use the same LR schedule, LR= $0.8$ initially and is decayed by $0.1$ at epoch 80 and again at epoch 250 with 300 epochs total budget. We measure $G_t$ and $N_t$ by averaging their values over the last 50 epochs of the second phase (i.e., from epoch 200 to 250).

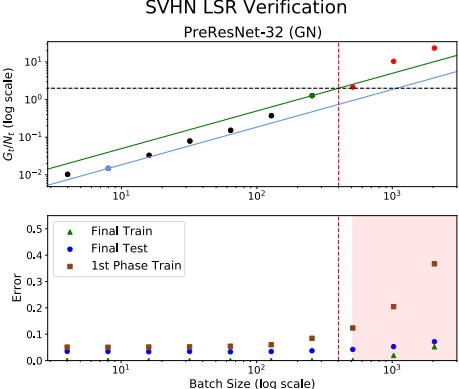

Figure 17: Further verification for our theory on predicting the failure of Linear Scaling Rule. We test if the condition applies to PreResNet-32 with GN trained on SVHN Netzer et al. (2011). LR= 0.8 initially and is decayed by 0.1 at epoch 100 with 120 epochs total budget. We measure $G_t$ and $N_t$ by averaging their values over the last 20 epochs of the second phase (i.e., from epoch 80 to 100).

### F.3 Additional Experiments for NGD (Noisy Gradient Descent)

We provide further evidence that SGD (1) and noisy gradient descent (NGD) (3) have similar train and test curves in Figures 18, 19, and 20. To perform NGD, we replace the SGD noise by Gaussian noise with the same covariance as was done in Wu et al. (2020). In Wu et al. (2020), the authors trained a network using BatchNorm, which prevents the covariance of NGD from being exactly equal to that of SGD. Hence, we use GroupNorm in our experiments, which improves NGD accuracy. We note that each step of NGD requires computing the full-batch gradient over the entire dataset (in this case, done through gradient accumulation), which is much more costly than a single SGD step. Each figure took roughly 7 days on a single RTX 2080 GPU.

**Explanation for the sudden drop of accuracy of GD in Figures 1, 19 and 20:** Note that the drop happens after the train accuracy reaches 100% for a while and that the high training accuracy (i.e., small loss) will lead to vanishing gradients. Our hypothesis is that after reaching full train accuracy for a while, the weight decay term dominates the dynamics and the smoothness (sharpness) of the loss function increases as the weight norm decays. This is because the scale invariance of the network implies that the smaller the weight norm is, the larger the smoothness (sharpness) is. (see property (3) of Lemma E.7). When the weight norm is small enough, the learning rate becomes larger than 2/smoothness (2/sharpness), and the dynamics of GD become unstable.

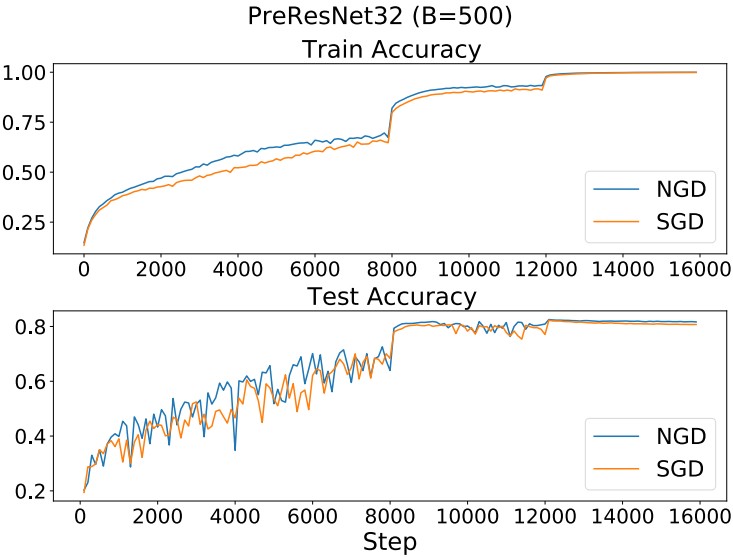

Figure 18: SGD and NGD with matching covariance have close train (top) and test (bottom) curves. The batch size for SGD is $500$ and LR= $3.2$ for both settings and decayed by $0.1$ at step $8000$. We smooth the training curve by dividing it into intervals of 100 steps and recording the average. For efficient sampling of Gaussian noise, we use GroupNorm instead of BatchNorm and turn off data augmentation. SGD and NGD achieve a maximum test accuracy of 82.3% and 82.5%, respectively

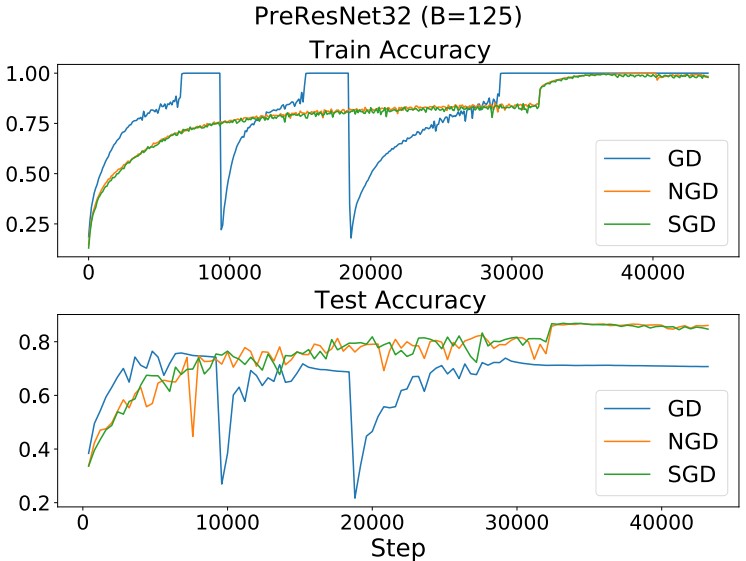

Figure 19: SGD and NGD with matching covariance have close train (top) and test (bottom) curves. The batch size for SGD is $125$ and LR= $0.8$ for all three settings and decayed by $0.1$ at step $32000$. We smooth the training curve by dividing it into intervals of 100 steps and recording the average. For efficient sampling of Gaussian noise, we use GroupNorm instead of BatchNorm and turn off data augmentation. GD achieves a maximum test accuracy of 76.5%, while SGD and NGD achieve 86.9% and 86.8%, respectively

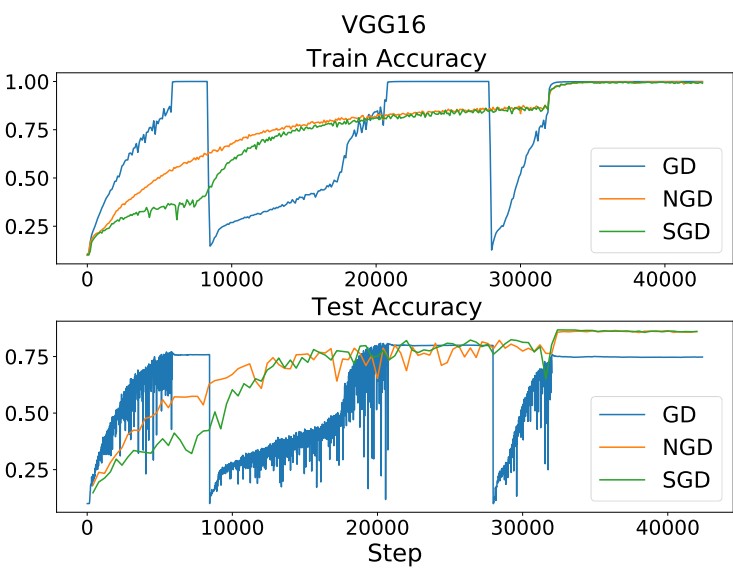

Figure 20: SGD and NGD with matching covariance have close train (top) and test (bottom) curves for VGG16. The batch size for SGD is $125$ and LR$= 0.8$ for all three settings and decayed by $0.1$ at step 32000. We smooth the training curve by dividing it into intervals of 100 steps and recording the average. For efficient sampling of Gaussian noise, we use GroupNorm instead of BatchNorm and turn off data augmentation. GD achieves a maximum test accuracy of 80.9%, while SGD and NGD achieve 86.8% and 86.5%, respectively