# OpenReview forum: "On the Validity of Modeling SGD with Stochastic Differential Equations (SDEs)"
_NeurIPS.cc/2021/Conference — NeurIPS 2021 Poster_

### Official Review · Reviewer_GfkN · 2021-07-10

**Rating:** 6
**Confidence:** 4

**Summary:**

SDE has been used as an approximation to help understand the implicit bias of SGD. This paper studies the validity of such approximations, which hasn't been formalized in prior work for realistic choices of learning rates. The contributions are three-folds:
  - Proposing "Stochastic Variance Amplified Gradient" (SVAG), an efficient numerical method to test whether the trajectories of SGD and the corresponding SDE are close.
  - Empirically verifying SVAG can be made to follow SGD. Since SVAG provably follows SDE under certain conditions, this means SDE also tracks SGD well by transitivity.
  - Providing necessary conditions for SDE and LSR to track SGD.


**Limitations And Societal Impact:**

The authors discuss implications of their theoretical results and potential insights for practitioners. Immediate societal impacts beyond the research community do not apply.

**Main Review:**

**Significance**: this work studies the importance question of the validity of approximating SGD with SDE. However, this paper does not provide a formal answer. Rather, the paper provides an indirect argument via the proposed SVAG: SVAG provably follows a certain SDE and empirically tracks SGD well, hence such SDE should also be close to SGD.

On the positive side, since SVAG can be made to track SGD well empirically, this suggests we can potentially study SVAG (or its approximations) instead of SGD. Moreover, the paper provides conditions for the applicability of LSR (i.e. NSR not too small, batch size not too large) which are valuable contributions.

However, the fact that SVAG interpolates between SGD (when taking $l=1$) and SDE (when taking $l \rightarrow \infty$) seems to be a negative result for using SDE to approximate SGD, since this means SDE and SGD sit on opposite sides of the spectrum and are _not_ close to each other. This suggests that SVAG is not the proper tool to study the SDE approximation to SGD, and I would like to have the authors' clarification on this.

**Writing**: the paper states the messages of this work are clear overall, but some
- Typo: end of line 65: "the benefit of SGD"
- Typo: end of line 266: "the equilibrium distributions"
- The first paragraph of Sec 5: what's the purpose of the discussion on weight decay?
- Fig 4: the figures are a bit too crowded to read; perhaps also consider changing the color scheme since currently it's hard to tell between $l=1$ vs $l=32$.
- Typo: end of line 828 (Thm E.1) should be a stop not a colon.

========
Post-rebuttal update: I thank the authors for their responses and have raised my score.

**Time Spent Reviewing:**

4.5

---

> ### Author Response · Authors · 2021-08-10
> **Author Response**
>
> Thank you for your efforts in the review process.
>
> **Response to “SDE is not close to SGD because they’re on the opposite sides of SVAG”**
>
> SVAG does indeed interpolate between SGD and the SDE approximation, but this does not mean that the two trajectories have to be far apart. In fact, as shown in Figure 4 (left) and many more settings in Section F.1, the SVAG trajectories for $l=1$ (i.e., SGD) and for larger $l$ (i.e., closer to the SDE) are very close to each other for many common architectures and datasets.
>
> In summary, our main argument proceeds as follows: SVAG provably converges to the SDE as $l\to\infty$ (Theorem 4.3), and it seems to converge for the values of $l$ in our experiments (Figure 4, Section F.1). These experiments also show that SVAG with large $l$ is close to the SGD (i.e., $l=1$) trajectory. Therefore, our theory and experiments suggest that the SDE is a good approximation of SGD. The final version will include this clarification.
>
> **Purpose of the discussion on the weight decay**
>
> The necessary condition in Sec 5 relies on the existence of equilibrium and thus on the usage of Weight Decay. As shown in [(Li et al., 2020)](https://arxiv.org/abs/2010.02916), weight norm converges to a stationary value for scale invariant nets when WD is turned on. On the other hand, without WD, the assumed equilibrium won’t even exist, because the gradient is always perpendicular to the weight itself, meaning the weight norm is strictly monotone increasing thanks to the Pythagorean Theorem. (see full argument in [(Arora et al., 2019b)](https://arxiv.org/abs/1812.03981))
>
> If our responses address your primary concerns, we would kindly ask you to adjust your review score. We are also happy to answer any additional questions in the open discussion period.

---

> > ### Comment · Reviewer_GfkN · 2021-08-22
> > **Clarification on Group Norm results**
> >
> > I thank the authors for the clarifications.
> >
> > Regarding experimental results on the closeness of SDE (approximated by $\ell = 32$) and SGD ($\ell = 1$), the curves are reasonably close when using batch norm but are quite different for group norm. Could you comment on why this is the case please?

---

> > > ### Author Response · Authors · 2021-08-23
> > > **Author Response**
> > >
> > > Indeed we don’t have any reason to expect that BN and GN will exhibit the same gap between l=1 (SGD) and l=32 (closer to the SDE). Using different normalization methods changes the architecture and could lead to very different performances in general. For example, it’s known that BN outperforms LN and GN on ResNet from the [Group Normalization paper](https://arxiv.org/pdf/1803.08494.pdf) but [other works](https://arxiv.org/pdf/2003.07845.pdf) show that LN works better for transformers. In our experiments, we also found that BN outperformed GN for SGD.
> > >
> > > Finally, we want to stress that though it’s completely not clear from theory what combination of dataset, architecture, learning rate, and batch size makes SGD close to SDE, our contribution is that we provide a simple and efficient way (SVAG) to simulate the SDE limit so we can at least empirically compare SGD to a high-resolution approximation for SDE on any given setting.

---

> > > > ### Comment · Reviewer_GfkN · 2021-08-29
> > > > **Thank you for your response**
> > > >
> > > > I thank the authors for the response. I will raise my score to 6.

---

### Official Review · Reviewer_aTSh · 2021-07-13

**Rating:** 8
**Confidence:** 3

**Summary:**

The paper has two main contributions. Firstly, it presents a practical simulation algorithm, SVAG, which is shown to converge weakly to the Itô SDE approximation of SGD over finite time horizons (with no restrictions on the learning rate $\eta$). Experimental evidence on deep nets is given to support the claim that SVAG can be used as a diagnostic for empirically testing if SGD and its SDE approximation have similar behaviour. The second main contribution provides sufficient conditions for the failure of the SDE approximation and linear scaling rule when the loss function is scale invariant. This theory is further supported by experiments on deep nets.

**Limitations And Societal Impact:**

* I believe the authors could better address the potential limitations of their experiments (see "Weaknesses" part of main review).

* Theorem 4.3 only applies on a finite time horizon $[0,T]$, whereas the optimization/MCMC literature is usually concerned with quantities obtained as $t\rightarrow\infty$. It could be worth highlighting this difference.

**Main Review:**

Strengths:

* The SVAG algorithm is new and interesting. It provides a practical methodology for weakly approximating the SDE corresponding to SGD, $\mathrm{d} X_t = -\nabla \mathcal{L}(X_t)\mathrm{d}t + \big(\eta \Sigma(X_t)\big)^\frac{1}{2}\mathrm{d}W_t$. To the best of my knowledge, it is the first algorithm that does so using stochastic gradients and with no restriction on $\eta$. The authors demonstrate how it can be applied as a diagnostic tool for testing if SGD and its SDE approximation have similar behaviour. This is useful as, for example, the linear scaling rule occurs when the SDE approximation holds (Smith et al., 2020).

* The paper gives new theoretical insights into the failure of the SDE approximation and linear scaling rule – which is certainly of practical importance. Their theory fits nicely with a wide array of experiments.

* The authors also discuss the role of non-Gaussian noise in their findings. In particular, they give experimental evidence that SGD and NGD can give similar test performances. This indicates that it is reasonable to model the gradient noise in SGD as Gaussian. Whilst these results are presented as secondary to the main results, they may be very relevant to the wider literature.

* The experiments are thorough.

* The paper is clearly written.

Weaknesses:

* The authors show that SVAG converges to the SDE in a weak sense, however in their experiment (figure 4), it appears they are making conclusions based on a single trajectory (which would only be justified by if SVAG converged to the SDE in a pathwise sense). If this is the case, then one or two sentences clarifying the limitations of the experiments would be helpful.

* Similarly, the authors claim that the convergence of SVAG to the SDE for "small values of $l$" is "confirmed in Figure 4". I find this to be too strong a claim and would prefer the statement to be weaker, for example "consistent with" instead of "confirmed".

Conclusion:

Overall, I believe this is a strong paper. In particular, it provides interesting theory and practical methodologies for testing the effectiveness of SDE approximations (which are already popular tools to studying SGD). The paper supports this with a wide variety of experiments.

Minor comments:

* In the introduction, the authors write "but the proof needs the LR of SGD to be an unrealistically small (unspecified) constant". How do we know if the LR is unrealistic if it is unspecified?

* The authors write "at equilibrium, the gradient norm must be smaller than its variance". Is it not the squared gradient norm?

* I'm not sure if plotting the train and test accuracies of standard gradient descent in Figure 3 is helpful.

* It is not clear (from the main body of text) why the scale invariance of the loss function $\mathcal{L}_\gamma$ is essential to the theory in Section 5.

Typos:

* Stochastic Gradient Gescent (page 1)
* $E\big[\big(\nabla \mathcal{L}_\gamma (X) -  \nabla \mathcal{L}(X))\big(\mathcal{L}_\gamma (X) -  \nabla \mathcal{L}(X)\big)^{\mathsf{T}}\big]$ is missing a $\nabla$ (pages 2 and 5)
* $\sqrt{\frac{\eta}{l}} \Sigma^l(x) = \sqrt{\eta}\Sigma^{1}(x)$ (page 5). Surely it is $\frac{\eta}{l} \Sigma^l(x) = \eta\Sigma^{1}(x)$.
* (ii) is used instead of (iii) in Lemma 4.5 (page 6)
* The left quotation mark in ”equilibrium” is the wrong way round (page 7)

**Time Spent Reviewing:**

3.5 hours

---

> ### Author Response · Authors · 2021-08-10
> **Author Response**
>
> Thank you very much for reviewing our paper. We really appreciate your positive reviews and insightful comments!
>
> **The authors show that SVAG converges to the SDE in a weak sense, however in their experiment (figure 4), it appears they are making conclusions based on a single trajectory (which would only be justified by if SVAG converged to the SDE in a pathwise sense). If this is the case, then one or two sentences clarifying the limitations of the experiments would be helpful.**
>
> You are correct that the experiments show only one trajectory and hence cannot fully verify the weak convergence we show theoretically. We will add a further discussion on the limitations of the experiments and amend the statement about SVAG converging for small values of l. We note that repeating our extensive experiments with multiple random seeds is quite computationally expensive, but we will do so for a few of the settings and include the results in the final version.
>
> **How do we know if the LR is unrealistic if it is unspecified?**
>
> In the final version we will clarify that the bound in (Li et al., 2019) needs the learning rate to be exponentially small in time, $e^{-CT}$, where C is not specified and depends on smoothness of the function, which is presumably large. Furthermore, note that they show SGD converges weakly to GF and SDE at the same rate. (indeed when LR goes to 0, SDE becomes GF) Thus whenever their requirements for LR are met, there should be no performance difference between SGD and full-batch GD (since they both converge to GF), which is not true in practice, e.g. see Figure 3. The final version will clarify this.
>
> **The meaning of the train and test accuracies of standard GD in Figure 3**
>
> We add full-batch GD as a baseline to rule out the possibility that NGD and SGD are close only because the noise is small and has little effect on the training. Through comparison between SGD and full-batch GD, it’s clear that SGD with standard LR is away from its deterministic limit, Gradient Flow.
>
> **It is not clear (from the main body of text) why the scale invariance of the loss function $L_\gamma$ is essential to the theory in Section 5.**
>
> On an intuitive level, scale invariance ensures that the gradient is orthogonal to the parameters, which removes confounding terms and gives us the simple equations at equilibrium, i.e. (23), (24) in the appendix. These two equations relate the three terms, norm square, gradient norm square and trace of gradient noise covariance in different ways, where the difference becomes remarkable and breaks LSI when learning rate is large or NSR is small. We will add more intuition about the importance of scale invariance in the future version.

---

### Official Review · Reviewer_3hdD · 2021-07-14

**Rating:** 7
**Confidence:** 4

**Summary:**

The authors consider a discrete time algorithm (SVAG) which is indexed by a parameter $l$ and which exactly corresponds to SGD when $l= 1$.  They show that this algorithm weakly converges to SGD's continuous SDE approximation as $l$ goes to $\infty$, and this without having to take the step-size infinitely small. They also provide a setting in which the SDE approximation provably fails to capture SGD's dynamics. Finally they give a testing rule for the SDE approximation to hold.

**Limitations And Societal Impact:**

Limitations are discussed.

**Main Review:**

The paper is well written and easy to read. The motivations underlying their work is clear and are pertinent to me. The related work is concise but relevant. All the results are clear and are discussed. My principle concern relies in the fact that the authors show that SVAG converges weakly to the classical SDE approximation (2), however (except for the experiments Figure 4)  it is not clear how SVAG and SGD compare as soon as $l$ is not equal to $1$. However I still believe that their paper gives some valuable insight on how and when the Itô SDE approximates well SGD for macroscopic step sizes.


Here are some minor concerns:
- l207: "(independent of $\eta$): I assume you mean "independent of $l$"
- maybe it is just me but it took me some time to understand that SVAG has the same first and second order moments as the Itô SDE (2) **when using a discretisation $\mathrm{d} t = \eta / l$** (and not $\mathrm{d} t = \eta$ as for SGD). This is clear when looking at the theorem but making it more explicit and explaining in the main text would make it easier to understand.
- Figure 1: what is the value of $l$ here ?
- Figure 3: how come there is this accuracy drop for gradient descent at step ~15000 ?
- Section 5.1: I would have enjoyed a a more in depth discussion around theorem 5.2: do you think this condition or a similar condition holds for other losses than scale invariant losses ? what is particular for scale invariant losses for this results to hold ? why consider a regularising term $\lambda \vert x \vert^2$, does the result still hold without regularisation ?
- Section 5: I am not familiar with scale invariant losses, my overall concern is how relevant are your results outside this class which (to me) seem very specific.

**Time Spent Reviewing:**

4

---

> ### Author Response · Authors · 2021-08-10
> **Author Response**
>
> Thanks a lot for your efforts in the review process. Below we answer your questions one by one.
>
> **My principal concern relies in the fact that the authors show that SVAG converges weakly to the classical SDE approximation (2), however (except for the experiments Figure 4) it is not clear how SVAG and SGD compare as soon as l is not equal to 1.**
>
> We kindly refer you to the extensive experiments in Appendix F.1 with different architectures, datasets, and learning rate schedules.
>
> **Sudden accuracy drop of GD in Figure 3**
>
> Note that the drop happens after the train accuracy reaches 100% for a while and that the high training accuracy (i.e., small loss) will lead to vanishing gradients. Our hypothesis is that after reaching full train accuracy for a while, the weight decay term dominates the dynamics and the smoothness (sharpness) of the loss function increases as the weight norm decays. This is because the scale invariance of the network implies that the smaller the weight norm is, the larger the smoothness (sharpness) is. (We only prove this for the gradient in Lemma E.5 in appendix, but the inverse scaling for hessian can be proved using the chain rule similarly). When the weight norm is small enough, the learning rate becomes larger than 2/smoothness (2/sharpness), and the dynamics of GD become unstable.
>
> **Why is scaling invariance needed in Sec 5?**
>
> On an intuitive level, scale invariance ensures that the gradient is orthogonal to the parameters, which removes confounding terms and gives us the simple equations at equilibrium, i.e. (23), (24) in the appendix. These two equations relate the three terms, norm square, gradient norm square and trace of gradient noise covariance in different ways, where the difference becomes remarkable and breaks LSI when learning rate is large or NSR is small.
>
> **Settings beyond scale invariant losses**
>
> We note that our necessary conditions on scale invariant parameters for LSI can be trivially extended to the case where the network is not entirely scale invariant, i.e., the last layer is not fixed. In that case, the gradient of scale invariant parameters (e.g., the parameters before normalization, the last layer excluded) is still perpendicular to its weight vector. Thus Eqn (23), (24) in appendix still hold, and the same conclusion follows, with the global norm (gradient, noise covariance) replaced by norm (gradient, noise covariance) of the scale invariant parameters.
>
> **Why is weight decay needed in Sec 5?**
>
> Weight Decay ($\ell_2$) is the one of the most popular regularizations used in deep learning and is also crucial to the result of Thm 5.2. Without WD, the assumed equilibrium won’t even exist, because the gradient is always perpendicular to the weight itself, meaning the weight norm is strictly monotone increasing thanks to the Pythagorean Theorem. (see [(Arora et al., 2019b)](https://arxiv.org/abs/1812.03981))

---

> > ### Comment · Reviewer_3hdD · 2021-08-23
> > **Thank you for the clarifications**
> >
> > Thank you for the clarifications. I believe this paper is well written and that the results make good progress towards understanding the validity of the SDE approximation for SGD. I choose to raise my score from 6 to 7.

---

### Official Review · Reviewer_Jauw · 2021-07-16

**Rating:** 8
**Confidence:** 3

**Summary:**

This paper validates the SDE approximation to SGD under finite (non-infinitesimal) learning rate by proposing a novel algorithm SVAG. This paper theoretically proves that SVAG is an order-1 weak approximation to the SDE, and empirical experiments show that SVAG behaves similarly to SGD. Using SVAG as a theoretical tool, this paper shows new theoretical insights in the linear scaling rule and the necessary condition for the SDE approximation.

**Limitations And Societal Impact:**

Limitations of the SVAG algorithm are adequately addressed.

**Main Review:**

-	It is claimed in this paper that the discrete nature of SGD is not an essential ingredient to the generalization (Line 130—133), as evidenced by Fig. 4. However, recently Smith et al. (2021, https://openreview.net/pdf?id=rq_Qr0c1Hyo) show a contradictory result that discretization influences generalization significantly by proving that discrete time SGD actually minimizes a “modified loss”. Admittedly, the difference in theory may originate from the different minibatch sampling mechanism considered in either paper. The mechanism considered in the current paper is random with replacement, while that in Smith et al. (2021) considers permutation of minibatches in each epoch (sample without replacement in each epoch), which is perhaps more practically relevant. However, I assume the empirical analysis in the current paper is performed with the usual minibatch sampling mechanism that is considered by Smith et al. (2021), but why do the empirical results in the current paper suggest that discretization is not essential? Suppose one considers the permutated minibatches like Smith et al. (2021) for SVAG, would the process that SVAG approximate becomes another continuous-time stochastic process but with the modified loss of Smith et al. (2021)? How can we reconcile the results in the current paper and Smith et al. (2021)?
-	In Fig. 4, SVAG generalizes better and is more stable than SGD as l increases at each effective step. Because SVAG is getting close to SDE as l increases, does it imply that the SDE would provide a good generalization performance given that we can exactly compute it? Would SDE yield a generalization upper bound for SGD?


**Time Spent Reviewing:**

4

---

> ### Author Response · Authors · 2021-08-10
> **Author Response**
>
> Thanks a lot for your positive and thoughtful feedback!
>
> **Reconciling our result with (Smith et al., 2021)**: Our conclusion that the discrete nature of SGD is not an essential ingredient for generalization performance doesn’t contradict with (Smith et al., 2021).
>
> SGD can have different continuous limits (e.g., gradient flow, SDE, etc) and we don’t claim that all such continuous limits demonstrate the same generalization performance as SGD does. (Smith et al., 2021) studies only deterministic flow with modified loss. Our point here is that there *exists* a continuous algorithm, the SDE approximation, which is the limit of SVAG as $l\to \infty$, that achieves similar performance or even slightly outperforms SGD. Therefore, to achieve good generalization, one doesn’t necessarily need to perform discrete updates, and any analysis that exclusively relies on the discrete nature of SGD is unlikely to capture the true phenomena responsible for the generalization mystery of deep learning.
> From a theoretical perspective, the main result of (Smith et al., 2021) doesn’t contradict SVAG in our setting. Their setting differs from ours in the minibatch sampling method, the time scaling, and the notion of approximation used.
> To be more specific, to simulate a step of SGD with LR $\eta\to 0$, SVAG is allowed $O(1/\eta) = \Theta(l)$ steps while (Smith et al., 2021) allows only constant number of steps, i.e., the ratio between the dataset size and the batch size. Our notion of closeness is defined in terms of weak approximation, i.e., close in distribution, while the notion of closeness of (Smith et al., 2021) is much weaker and only ensures the expectation of final updates are close. Therefore, their theoretical result doesn’t have any direct implications for our setting.
>
> **”Suppose one considers the permuted minibatches like Smith et al. (2021) for SVAG, would the process that SVAG approximate becomes another continuous-time stochastic process but with the modified loss of Smith et al. (2021)?”**:
>
> We thank the reviewer for bringing up this interesting and thoughtful question. It is beyond the scope of the current work and is left as future work. Our best guess is that the continuous limit for SVAG could be different from the standard SDE, but we doubt that it becomes a deterministic flow with the modified loss of Smith et al. (2021). This is because for SVAG with $l\to\infty$ (i.e., as SVAG approaches the SDE), the learning rate will go to 0 and the correction term in Smith et al. (2021) will disappear.
>
> From an experimental perspective, we provide a more detailed discussion of different sampling methods in Appendix F.1.1. In Figure 7, we show that SVAG has similar train/test curves regardless of which sampling method is used. We also want to stress that the **random shuffling** setting considered in Smith et al. (2021) is still not close to practice, because it ignores two tricks commonly used during training: 1. Data augmentation (like image horizontal flipping or cropping); 2. Batch Normalization. These tricks break the most important property of random shuffling SGD used in the analysis of Smith et al., (2021), that the average of batch loss functions in an epoch is equal to the expected loss function. Since all of our SVAG experiments use data augmentation, the results in Smith et al. (2021) don’t apply to our experimental setting.
>
>
> **Because SVAG is getting close to SDE as l increases, does it imply that the SDE would provide a good generalization performance given that we can exactly compute it? Would SDE yield a generalization upper bound for SGD?**
>
> Indeed, in our experiments, we observe that when SVAG converges to an SDE that is far from SGD, the SVAG limiting trajectory has better generalization performance. More experiments are needed to investigate whether the same phenomena holds for different datasets and architectures. Understanding the generalization properties of different training algorithms is in general a difficult open question, so we cannot draw any rigorous conclusions here.

---

> > ### Comment · Reviewer_Jauw · 2021-08-11
> > **A follow-up question**
> >
> > I thank the authors for a very insightful feedback and for pointing out the data augmentation used in experiments which my review has overlooked. You note that “since all of our SVAG experiments use data augmentation, the results in Smith et al. (2021) don’t apply to our experimental setting”. Does this statement imply the theoretical results in this paper could hold for the SGD with the minibatch reshuffling when the iid data augmentation is present (I suppose this is the setting under which the empirical results were obtained)? Minibatch reshuffling with iid data augmentation appears to deviate from the setup of Eq. (1). For example, if we consider adding iid mean 0 Gaussian noise to each pixel of the images in each minibatch, the augmented minibatches are not iid, and thus violating Eq. (1).

---

> > > ### Author Response · Authors · 2021-08-11
> > > **Response to the follow-up question**
> > >
> > > No, we did not mean to imply that our results hold for SGD with minibatch shuffling and iid data augmentation. As mentioned earlier, we leave the SVAG limit of it as a future question.
> > > However, as shown in Appendix F.1.1 and Figure 7, the three standard sampling methods (random shuffling, iid batch with replacement, iid batch without replacement) give almost the same train/test curves for SVAG with l=1,4,16, and therefore we believe the different sampling methods only have negligible effect in standard settings. Since the SVAG experiments are expansive (a single run of l=32 takes more than 96 gpu hours), we decided only to run the experiments with the most popular sampling method — random shuffling, which is also the default of Pytorch.

---

> > > > ### Comment · Reviewer_Jauw · 2021-08-23
> > > > **Thank you for the clarifications**
> > > >
> > > > Thanks for pointing out the results in Appendix which I have overlooked. I believe this paper has made a good progress toward understanding the dynamics of SGD, although obviously there are still many open problems. I would like to raise my rating from 7 to 8.

---

### Decision · Program_Chairs · 2021-09-27

**Decision:**

Accept (Poster)

**Comment:**

The paper proposes a practical algorithm called SVAG, which is shown to converge in the weak sense
to the SDE approximation of SGD, for even moderately large step sizes.
Authors demonstrate its performance via experiments on neural networks, and show that SVAG can diagnose if SGD and its SDE approximation have similar behaviors. Authors further investigate the conditions in which the SDE approximation fails.

The motivation of this paper is clear, and very timely. All the assumptions and the results are presented in a clear way.

All reviewers agree that this paper is well-written and its results are interesting. Authors should address reviewers' suggestions in the final version of their work. Specifically, authors should add a discussion on the limitations of the experiments and amend the statement about SVAG converging for small values of l, clarify the bound in Li et al., 2019, and add more intuition about the importance of scale invariance.